# A dual role for the N-terminal domain of the IL-3 receptor in cell signalling

Sophie E. Broughton[1], Timothy R. Hercus[2], Tracy L. Nero[1], Winnie L. Kan[2], Emma F. Barry[2], Mara Dottore[2], Karen S. Cheung Tung Shing[1,3], Craig J. Morton[1], Urmi Dhagat[1], Matthew P. Hardy[4], Nicholas J. Wilson[4], Matthew T. Downton[5], Christine Schieber[5], Timothy P. Hughes[6], Angel F. Lopez[2] & Michael W. Parker [1,3]

The interleukin-3 (IL-3) receptor is a cell-surface heterodimer that links the haemopoietic, vascular and immune systems and is overexpressed in acute and chronic myeloid leukaemia progenitor cells. It belongs to the type I cytokine receptor family in which the α-subunits consist of two fibronectin III-like domains that bind cytokine, and a third, evolutionarily unrelated and topologically conserved, N-terminal domain (NTD) with unknown function. Here we show by crystallography that, while the NTD of IL3Rα is highly mobile in the presence of IL-3, it becomes surprisingly rigid in the presence of IL-3 K116W. Mutagenesis, biochemical and functional studies show that the NTD of IL3Rα regulates IL-3 binding and signalling and reveal an unexpected role in preventing spontaneous receptor dimerisation. Our work identifies a dual role for the NTD in this cytokine receptor family, protecting against inappropriate signalling and dynamically regulating cytokine receptor binding and function.

[1] Australian Cancer Research Foundation Rational Drug Discovery Centre, St. Vincent's Institute of Medical Research, Fitzroy, VIC 3065, Australia. [2] The Centre for Cancer Biology, SA Pathology and the University of South Australia, Adelaide, SA 5000, Australia. [3] Department of Biochemistry and Molecular Biology, Bio21 Molecular Science and Biotechnology Institute, University of Melbourne, Parkville, VIC 3010, Australia. [4] CSL Limited, Parkville, VIC 3010, Australia. [5] IBM Research Australia, Level 5, 204 Lygon Street, Carlton, VIC 3053, Australia. [6] South Australian Health and Medical Research Institute and University of Adelaide, Adelaide, SA 5000, Australia. Sophie E. Broughton, Timothy R. Hercus and Tracy L. Nero contributed equally to this work. Angel F. Lopez and Michael W. Parker jointly supervised this work. Correspondence and requests for materials should be addressed to A.F.L. (email: angel.lopez@sa.gov.au) or to M.W.P. (email: mparker@svi.edu.au)

nterleukin (IL)-3 is a tightly regulated pleiotropic cytokine produced mainly by activated T lymphocytes that stimulates the production and function of multiple haematopoietic cell types as well as cells involved in the immune response such as dendritic cells[1,2]. Beyond haemopoiesis and immunity, IL-3 has also been shown to play a role in other biological systems such as angiogenesis[3] and in the development of the central nervous system[4,5]. In recent years, the IL-3 receptor system has come under the spotlight because stem/progenitor cells from patients with acute myeloid leukaemia (AML) overexpress the IL-3 receptor α-subunit (IL3Rα) and this is associated with reduced patient survival[3,6–9]. IL3Rα is also overexpressed in chronic myeloid leukaemia (CML) providing a promising target for therapy[10]. Hence, there are ongoing efforts to understand how the IL-3 receptor signals and to develop new therapies in AML and CML based on appropriately targeting IL3Rα.

IL-3 is a member of the beta common (βc) cytokine family, which also includes granulocyte-macrophage colony-stimulating factor (GM-CSF) and IL-5. These cytokines signal through heterodimeric cell-surface receptors that are expressed at low levels and comprise a cytokine-specific α-subunit and the shared βc subunit[1]. Activation of the IL-3 receptor is thought to involve sequential assembly of a receptor signalling complex whereby the critical step is the initial binding of IL-3 to IL3Rα[2,11–13], followed by recruitment of βc and the assembly of a high order complex which, by analogy with the GM-CSF receptor[14], would bring JAK2 molecules together to trigger downstream signalling. Interestingly, the initial binding of the βc cytokines to the α-subunits is with low affinity that varies widely among the three receptors, yet the high-affinity binding achieved when βc is present is the same ($K_D$ ~200 pM) for all three cytokines. IL-3 is remarkable because of its very low affinity for IL3Rα ($K_D$ ~100–200 nM), much lower than the affinity of GM-CSF ($K_D$ 2–10 nM) or IL-5 ($K_D$ 1–2 nM) for their respective α-subunits. However, it is not known how IL3Rα recognises IL-3 nor how the IL-3 binary complex overcomes such a low affinity interaction and yet achieves the same high affinity as GM-CSF and IL-5 in the presence of βc. Previous studies have demonstrated that IL-3 K116 variants, in particular the K116W variant, have higher affinity for IL3Rα compared to wild-type IL-3 and enhanced proliferative activity[15–17]. Mutant cytokines with higher affinity for their receptors and/or higher functional potency have been dubbed 'superkines'[18]. A molecular explanation for how a single mutation converts IL-3 into a superkine has not been forthcoming.

We have recently solved the structure of the extracellular region of IL3Rα bound to the blocking antibody CSL362 that is currently in clinical trials for AML[11]. The extracellular region of IL3Rα comprises three fibronectin type III (FnIII) domains: an Ig-like N-terminal FnIII domain (NTD) followed by a pair of FnIII domains: domain 2 (D2) and domain 3 (D3). Interestingly, two CSL362:IL3Rα complexes were observed, very similar in their overall molecular arrangements, except for the orientation of the IL3Rα NTD. The crystals revealed two distinct IL3Rα conformations, an 'open' form in which the angle between the NTD and D2 domains of IL3Rα is ~95°, and a 'closed' form in which the angle between the NTD and D2 domains is ~55° and is comparable to the 'wrench-like' conformation[11] reported for the related cytokine receptors GMRα[19], IL5Rα[20,21], IL13Rα1[22] and IL13Rα2[23]. The reason for these two conformations is unclear but it has been hypothesised that the NTD of IL3Rα, like other members of the type I cytokine receptor family, may provide molecular flexibility to facilitate cytokine recognition and allow appropriate presentation of cytokine to the signalling subunits of these receptor families. Intriguingly, the NTD is omitted in one isoform of IL3Rα (IL3Rα SP2) and although not essential, the NTD contributes significantly to wild-type IL-3 function[24].

Here we present the crystal structures of the IL3Rα extracellular domains bound to wild-type IL-3 and to the IL-3 K116W superkine. The structures show the detailed molecular interfaces between IL3Rα and ligand and, in particular the mechanism of IL3Rα NTD engagement with the cytokine. All-atom explicit-solvent molecular dynamics (MD) simulations reveal that the IL3Rα NTD is highly mobile and remains so when bound to wild-type cytokine, but becomes relatively rigid when bound to IL-3 K116W, indicating that the NTD interaction can be distally regulated. Structure-guided mutagenesis, along with functional and biochemical studies, establish a critical role for the IL3Rα NTD in optimising IL-3 binding and function and in preventing spontaneous receptor heterodimerisation with βc. Given their overall structural conservation, the NTD may perform similar roles in other members of the type I cytokine receptor family.

## Results

**IL-3 receptor recognises cytokine differently.** We have determined the crystal structure of IL-3 bound to its receptor α-subunit, IL3Rα, to 2.7-Å resolution (Methods, Table 1, Fig. 1, Supplementary Fig. 1). Two complexes were found in the asymmetric unit and they overlay closely (Supplementary Fig. 2a) with root-mean-square deviation (RMSD) of the Cα atoms of 0.7 Å (additional detail in Supplementary Note 1). IL3Rα residues P25 to F96 define the Ig-like NTD, residues E108 to F202 define D2 and the third FnIII domain, D3, consists of residues T209 to D294. D2 and D3 constitute the cytokine recognition module (CRM) (Fig. 1a). IL3Rα contains N-linked glycans at N80, N109 and N218 (Fig. 1a) that are required for optimal IL3Rα expression (Supplementary Fig. 3a, b and see additional detail in Supplementary Note 1).

### Table 1 Data collection and refinement statistics

| | Wild-type IL-3 binary complex | IL-3 K116W binary complex |
|---|---|---|
| Data collection | | |
| Space group | $P6_522$ | $P6_1$ |
| Cell dimensions | | |
| $a, b, c$ (Å) | 132.0, 132.0, 210.6 | 106.5, 106.5, 96.1 |
| $\alpha, \beta, \gamma$ (°) | 90, 90, 120 | 90, 90, 120 |
| Resolution (Å) | 48.1–2.7 (2.83–2.7) | 48.1–2.4 (2.40–2.39) |
| $R_{meas}$ | 0.19 (0.83) | 0.08 (0.69) |
| $R_{pim}$ | 0.09 (0.39) | 0.02 (0.17) |
| $I/\sigma_I$ | 25.7 (4.8) | 19.7 (3.4) |
| CC1/2 | 0.99 (0.81) | 0.99 (0.74) |
| Completeness (%) | 100 (100) | 99.5 (97.5) |
| Redundancy | 8.1 (8.5) | 16.1 (16.1) |
| Refinement | | |
| No. of reflections | 13,304 | 25,439 |
| $R_{work}/R_{free}$ (%) | 22.6/28.9 | 21.6/24.2 |
| No. of atoms | | |
| Protein | 5966 | 5706 |
| Water | 100 | 58 |
| B-factors (Å²) | | |
| Main chain | 44.5 | 70.0 |
| Side chain | 45.2 | 72.7 |
| Ligand/ion | 67.6 | 90.0 |
| Water | 32.7 | 59.3 |
| RMSD | | |
| Bond lengths (Å) | 0.02 | 0.03 |
| Bond angles (°) | 2.3 | 2.5 |

Data were collected from a single crystal per complex. Values in parentheses are for highest resolution shell. Related to Figs. 1, 2, 4, 5, Supplementary Figs. 1, 2 and 4

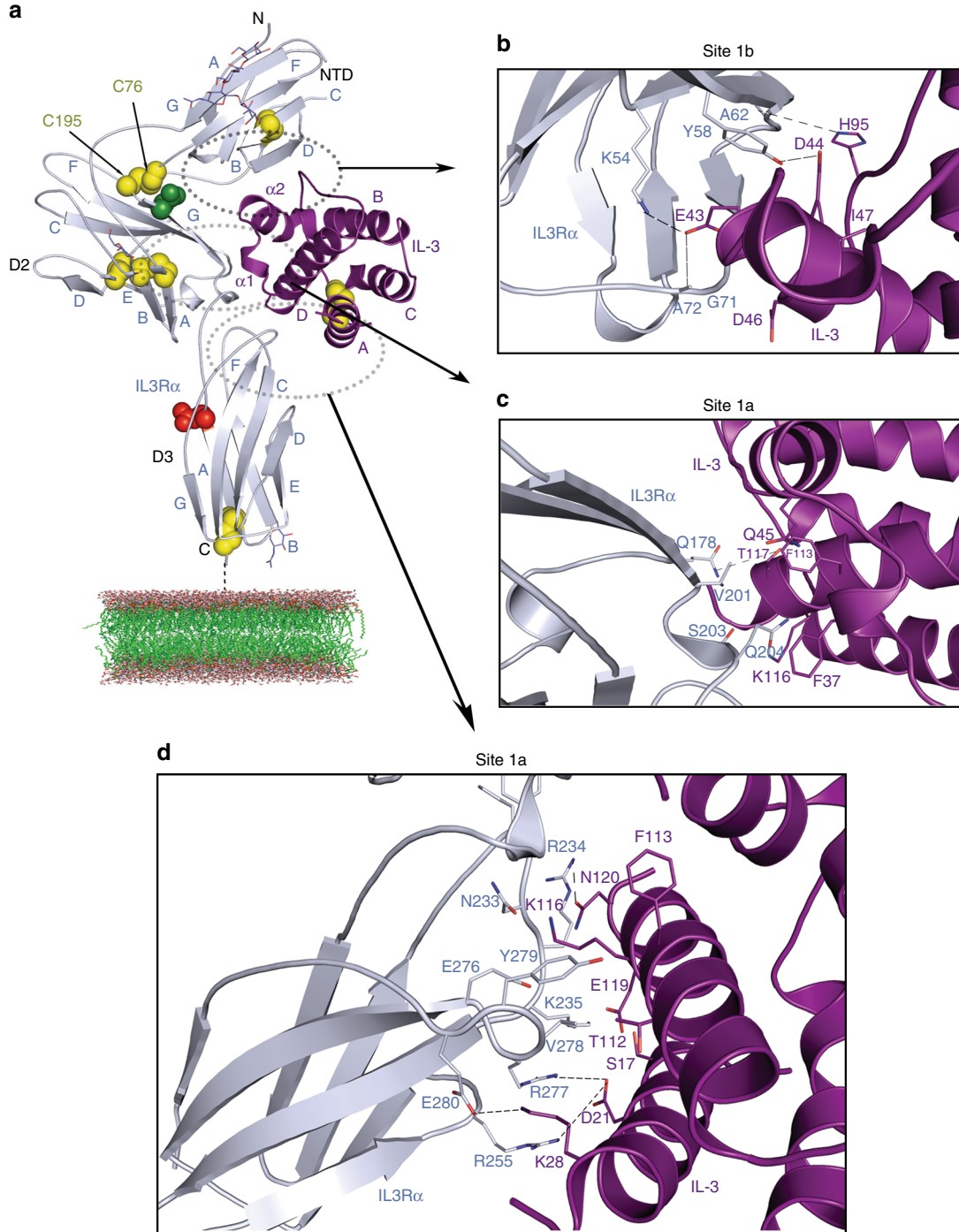

**Fig. 1** Crystal structure of the IL-3 binary complex. **a** The structure of IL3Rα (coloured blue-grey in all panels) bound to the four α-helical bundle of IL-3 (coloured purple in all panels). The β-strands that make up the three FnIII domains (NTD, D2 and D3) and the N- and C-termini are labelled. The cysteine residues in IL3Rα are represented as yellow spheres and the bound sugars at N80, N109 and N218 of ILRα are shown as sticks. The mutated N212Q residue is shown as red spheres. D197 is represented as green spheres. **b** A close-up view highlighting key interactions between IL-3 and the IL3Rα NTD (site 1b). Hydrogen bonds are shown as black dashed lines in all panels. **c** A close-up view highlighting key interactions between IL-3 and the IL3Rα D2–D3 linker region (part of site 1a). **d** A close-up view highlighting key interactions between IL-3 and the IL3Rα D3 (part of site 1a). Together, **c** and **d** define the complete site 1a

The three IL3Rα FnIII domains adopt a 'wrench-like' conformation that wraps around the four helical bundle of IL-3. The hinge region between the D2 and D3 domains has an elbow angle of ~100°, which is within the range observed in related type I cytokine receptor structures[11,19–23]. The elbow angle between D2 and NTD is ~65° and a disulphide bond (C76–C195) provides an additional covalent interaction between

these domains that is not found in related receptors. The receptor component of the IL-3:IL3Rα complex is similar to the IL3Rα structures previously determined in complex with an antibody[11]; the CRMs are essentially identical (RMSD over the Cα atoms is 0.5 Å); however, the angle of the NTD with respect to the CRM in the IL-3:IL3Rα complex lies between the angles observed in the open and closed forms of the published

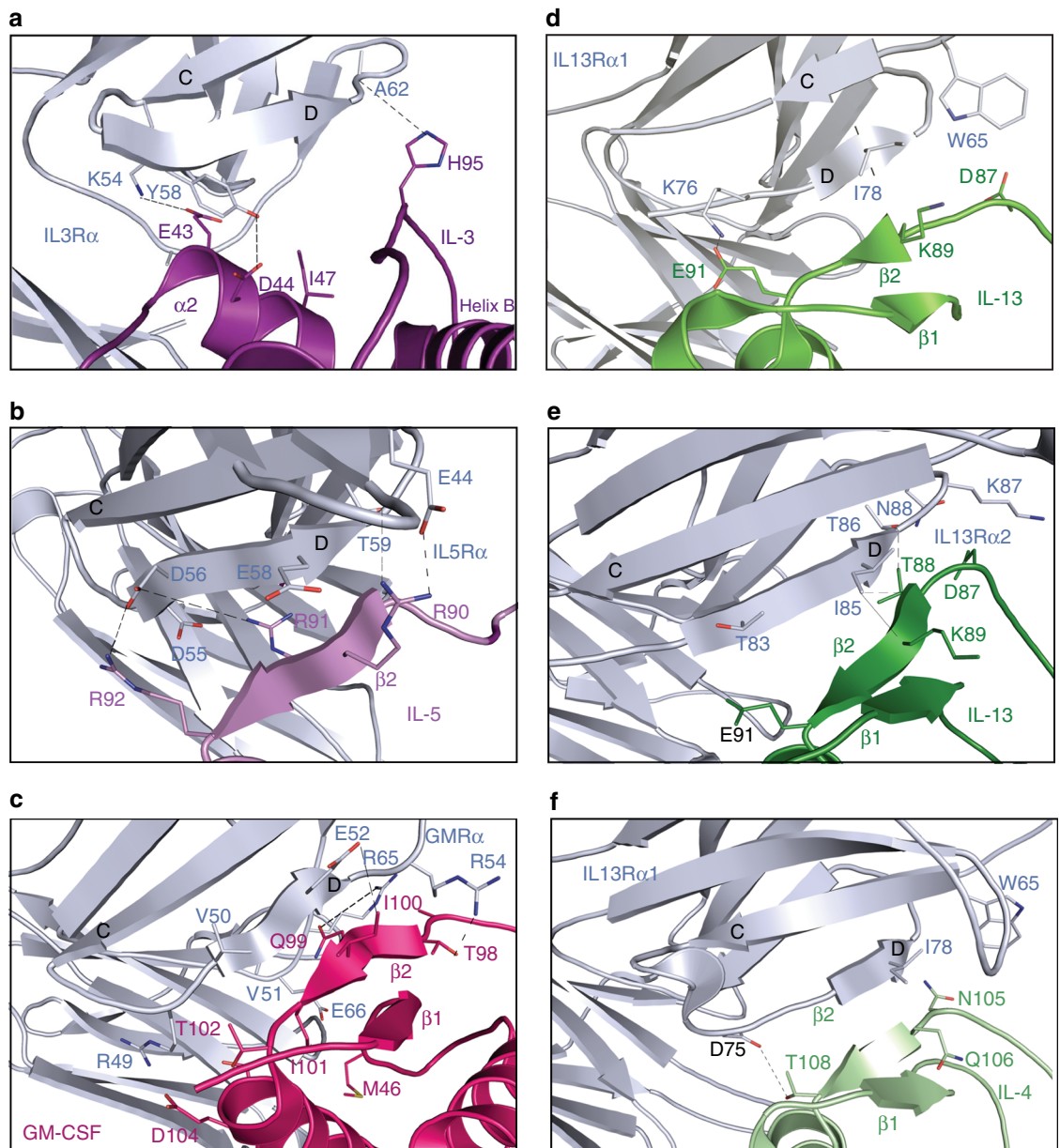

**Fig. 2** Comparison of IL3Rα site 1b interactions to related cytokine receptor complexes. **a** Wild-type IL-3:IL3Rα binary complex. **b** IL-5:IL5Rα binary complex (PDB ID: 3VA2, 3QT2[20, 21]). **c** GM-CSF:GMRα binary complex (PDB ID: 4RS1[19]). **d** IL-13:IL13Rα1 component of the ternary complex (PDB ID: 3BPO[22]). **e** IL-13:IL13Rα2 binary complex (PDB ID: 3LB6[23]). **f** IL-4:IL13Rα1 component of the ternary complex (PDB ID: 3BPN[22])

structures (Supplementary Fig. 2b and additional detail in Supplementary Note 1).

IL3Rα contacts IL-3 through an interface containing two distinct interaction sites, site 1a and site 1b. The entire interface consists of 12 hydrogen bonds, 4 salt bridges and 16 residues involved in van de Waals interactions; a complete list of these interactions are detailed in Supplementary Table 1. Site 1a comprises the interactions between cytokine and the IL3Rα CRM and corresponds to the classical site 1 of simpler type I homodimeric cytokine receptors[25]. The EF loop in D2, the D2–D3 linker region and the BC and FG loops of D3 define the IL3Rα contribution to site 1a (Fig. 1c, d, Supplementary Fig. 1c). Site 1b, the specific molecular interaction between cytokine and the IL3Rα NTD, is unique among the βc and IL-13 receptors and comprises the C-, D- and E-strands and the DE and EF loops of the IL3Rα NTD (Fig. 1b) that interact with IL-3 via the α2-helix, and the CD loop (Fig. 1a, Supplementary Figs. 1c, 4a).

The most surprising difference between the IL-3 binary complex and related cytokine complexes is the manner in which IL-3 interacts at site 1b. In all related receptor complexes, the cytokine interacts with the receptor subunit NTD via a β-sheet hydrogen bonding network between the D-strand of the NTD and the β2-strand of the cytokine (Fig. 2, Supplementary Fig. 4a and additional detail in Supplementary Note 1). Unlike GM-CSF, IL-4, IL-5 or IL-13, IL-3 does not have a structural equivalent to the β2-strand (Fig. 2, Supplementary Fig. 4a). In the IL-3 binary complex, the D-strand of the IL3Rα NTD is closest to the two-turn IL-3 α2-helix (residues N41-M49, located on the AB loop) and the CD loop. IL-3 makes only five polar interactions with the IL3Rα NTD; one between the side chain of IL-3 H95 and the backbone carbonyl of IL3Rα A62 and a second between IL-3 D44 (located on the α2-helix) and the hydroxyl group of IL3Rα Y58 (Figs. 1b, 2a). In addition, a network of three polar interactions between IL-3 E43 and IL3Rα K54, G71 and A72 (Figs. 1b, 2a)

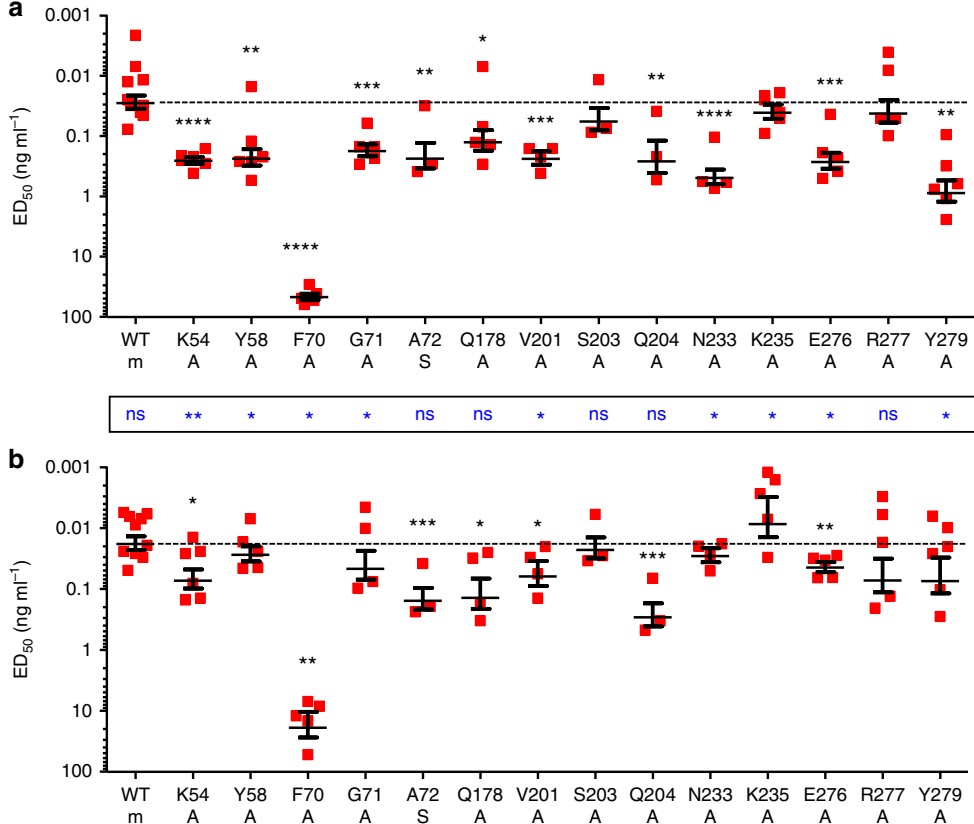

**Fig. 3** IL3Rα site 1b residues are a functional hotspot in mediating IL-3 signalling. Functional activity of IL3Rα mutants was measured in **a** wild-type IL-3 or **b** IL-3 K116W dose–response proliferation assays using CTLL-2 cell lines transduced to co-express the βc subunit with either wild-type or mutant forms of IL3Rα. The identity of each IL3Rα mutant is shown along with the substituted residue (m). Half-maximal responses (ED$_{50}$) were calculated from each experiment and averaged. The dashed line represents the response from cells expressing wild-type IL3Rα. Errors represent SEM. Statistical significance of differences in functional response between wild-type IL3Rα and the mutant forms of IL3Rα ($p$) was determined using a two-tailed unpaired $t$ test and are shown as black asterisks above the data for each mutant. Statistical significance of differences in functional response between wild-type IL-3 and IL-3 K116W ($p$) was also determined using a two-tailed unpaired $t$ test and are shown as blue asterisks between data sets **a** and **b**. ns $p > 0.05$, *$p < 0.05$, **$p < 0.01$, ***$p < 0.001$, ****$p < 0.0001$

further stabilises the IL-3:IL3Rα NTD (site 1b) interaction. The IL-3 α2-helix tucks into a crevice between the D-strand and the EF loop of the IL3Rα NTD and it is a unique point of stabilisation that does not occur in GM-CSF, IL-4, IL-5 or IL-13 (Fig. 2, Supplementary Fig. 4a). A detailed analysis of the site 1a and site 1b interaction surfaces in the IL-3:IL3Rα binary complex structure shows that multiple IL-3 residues contribute to IL3Rα binding (Supplementary Table 1).

**NTD contributes to optimal IL-3 binding and signalling.** We have recently identified functional roles for a distinct set of IL3Rα residues in mediating direct IL-3 binding and from our crystal structure we observe that these residues are distributed across both site 1a and site 1b (Fig. 1, Supplementary Table 1)[11]. In site 1a, Q178 in the EF loop of D2, V201 and S203 in the D2/D2–D3 linker region, N233 in the BC loop of D3 and E276, R277 and Y279 in the F-strand/FG loop of D3—all have a functional role. In site 1b, the key functional residues are K54 in the C-strand, Y58 in the D-strand and G71 and A72 in the EF loop of the NTD.

We previously used surface plasmon resonance and the soluble form of the α-subunit (ie, extracellular domain only, sIL3Rα) to investigate the contribution of these IL3Rα residues to IL-3 binding[11]. Compared to wild-type sIL3Rα ($K_D = 220$ nM), alanine mutations of Y58, G71 in the NTD, and E276 or Y279 in site 1a

abolished measurable IL-3 binding ($K_D > 5000$ nM), while alanine mutations of site 1a Q178, V201, S203, N233 and R277 produced modest reductions in IL-3-binding affinity ($K_D = 580–2250$ nM). These differences in IL-3 binding became masked when full-length receptor mutants were co-expressed with βc on COS cells and wild-type high-affinity binding was restored[11] while also demonstrating that the loss of measurable IL-3 binding does not neccessarily mean abolition of IL-3 interaction. Despite this, CTLL-2 cells co-expressing βc and many of the IL3Rα mutants (Supplementary Fig. 5) are significantly less responsive to IL-3 stimulation in cell proliferation studies than cell expressing wild-type IL3Rα (Fig. 3, Supplementary Table 2). In particular, mutation of residues K54, Y58, F70, G71 and A72 in the NTD all led to significant reductions in IL-3 function (ED$_{50} = 0.18–46.6$ ng ml$^{-1}$) compared to wild-type IL3Rα (ED$_{50} = 0.028$ ng ml$^{-1}$). In the IL3Rα CRM region, only mutation of Q178, V201, Q204, N233, E276 and Y279 led to a significant reduction in IL-3 function (Fig. 3, Supplementary Table 2). The reduction in IL-3 signalling we observe for these IL3Rα mutations may arise as a result of reduced IL-3 binding to IL3Rα, but may also reflect subtle alterations in the engagement of βc in IL-3 signalling complexes utilising mutant IL3Rα. Mutation of the buried F70 residue has a drastic effect on IL-3 responsiveness (ED$_{50} = 46.6$ ng ml$^{-1}$); however, this may be attributed to a major reduction in the expression of the IL3Rα F70A mutant (Supplementary Fig. 5).

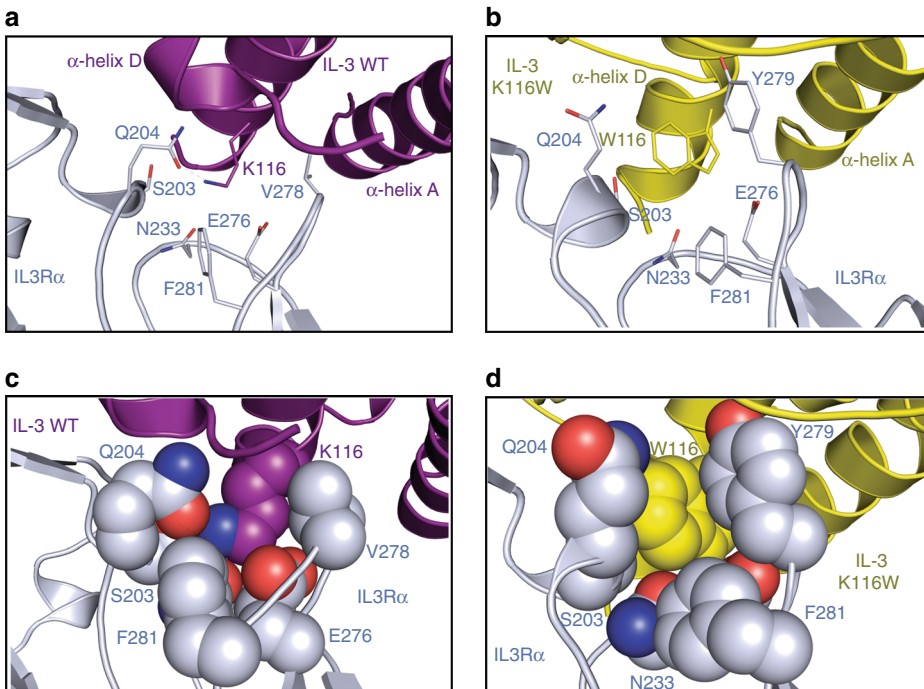

**Fig. 4** Structure of IL3Rα bound to IL-3 K116W. **a** Close-up view of the environment around IL-3 K116 in site 1a of the wild-type IL-3 binary complex. Colour scheme as described for Fig. 1b–d. **b** Close-up view of the environment around W116 in site 1a of the IL-3 K116W binary complex. IL3Rα coloured blue-grey and IL-3 K116W coloured yellow. **c** Residues around IL-3 K116 in the wild-type IL-3 binary complex displayed as spheres to demonstrate the volume they occupy and the residue packing at this site. **d** As for **c**, but for the IL-3 K116W binary complex

These experiments identify multiple residues in IL3Rα that are distributed across site 1a and site 1b, which contribute significantly to IL-3 function.

**IL3Rα recognises IL-3 K116W via a network of interactions.** To further explore the role of site 1 in IL-3 binding and signalling, we determined the crystal structure of IL-3 K116W bound to IL3Rα at 2.4-Å resolution (Methods, Table 1, Supplementary Fig. 1). The IL-3 and IL-3 K116W binary complex structures are essentially identical (Supplementary Fig. 2b) with an overall RMSD over the Cα atoms of 0.6 Å (RMSD for IL3Rα is 0.5 Å and for IL-3 is 0.3 Å). The cytokine buried surface area is similar in both complexes (1150 Å$^2$ for IL-3 vs 1166 Å$^2$ for IL-3 K116W), with close surface complementarity values ($S_c = 0.69$ for IL-3 vs 0.71 for IL-3 K116W) and a similar number of polar interactions between the cytokine and IL3Rα (11 hydrogen bonds (one less than wild type), 5 salt bridges (one more than wild type)).

Not surprisingly, the most significant differences between the wild-type IL-3 and IL-3 K116W structures occur within the vicinity of residue 116 (Fig. 4). Within the cytokine itself, the single-turn α1-helix located on the wild-type IL-3 AB loop unfurls slightly in IL-3 K116W (Supplementary Fig. 4b, d) to allow F37 to interact via edge-to-face π–π interactions with W116, which in turn interacts with F113 (also via edge-to-face π–π interactions). The π–π interaction network formed between F37-W116-F113 stabilises the cytokine AB loop and reduces the loop's overall mobility as assessed by comparisons of the Cα RMSD for the cytokines alone (200 ns MD simulation, Supplementary Fig. 4b, c) and Cα contact distances between F37, F113 and K/W116 in the binary complexes (100 ns MD simulations, Supplementary Fig. 7b, c) (additional detail in Supplementary Note 1). In the wild-type IL-3 binary complex, the IL-3 K116 side chain points directly into an IL3Rα hydrophobic pocket lined by residues Q204, N233, V278, Y279 and F281, and is further capped by F37 in IL-3 (Fig. 4a, c). The IL-3 K116 side chain is stabilised by polar interactions with S203,

Q204 and N233 in IL3Rα, and N120 in IL-3. In contrast, the W116 side chain of IL-3 K116W slots into a groove formed by Q204 and Y279 of IL3Rα, and forms edge-to-face π–π interactions with F37 and F113 in IL-3, and Y279 and F281 in IL3Rα. This results in the IL-3 K116W side chain sitting snuggly in a site 1a pocket formed by F37, T112, F113, T117, E119 and N120 of IL-3 and S203, Q204, N233, E276, V278 and Y279 of IL3Rα (Fig. 4b, d). Additional interactions with the IL-3 K116W side chain also have a downstream effect on the adjacent C-terminal end of helix D in IL-3, which is involved in van der Waals contacts with IL3Rα via residues E119, A121 and A123.

**Differing effects of site 1 residues on IL-3 K116W function.** We assessed the impact of IL3Rα site 1 mutations on IL-3 K116W function using the CTLL-2 cell lines co-expressing βc and IL3Rα mutants (Fig. 3, Supplementary Table 2). In general, mutations of IL3Rα site 1 residues had less impact on IL-3 K116W function than for wild-type IL-3. We observed that mutation of residues K54, A72, Q178, V201, Q204 and E276A led to significant reductions in IL-3 K116W function ($ED_{50} = 0.044$–$0.29$ ng ml$^{-1}$) compared to wild-type IL3Rα ($ED_{50} = 0.018$ ng ml$^{-1}$). As for wild-type IL-3, the drastic impact of the IL3Rα F70A mutation on IL-3 K116W responsiveness ($ED_{50} = 18.8$ ng ml$^{-1}$) can be largely attributed to reduced IL3Rα F70A expression levels (Fig. 3, Supplementary Fig. 5, Supplementary Table 2).

IL3Rα S203, Q204, N233 and Y279 all interact directly with both K116 of IL-3 and W116 of IL-3 K116W (Fig. 4, Supplementary Table 1). Interestingly, both cytokines display significantly reduced functional activity on CTLL-2 cells expressing the IL3Rα Q204A mutant (Fig. 3, Supplementary Table 2). In contrast to Q204, mutation of S203 does not affect the function of either cytokine while mutation of N233 or Y279 significantly reduces the function of wild-type IL-3 only suggesting that these residues directly influence the K116/W116 cytokine interaction. Mutation of IL3Rα residues in the vicinity of the IL-3 K116/W116

**Table 2 Enhanced site 1a interactions do not fully compensate for the loss of site 1b binding interactions through the NTD**

| Cell | Receptor subunits | Wild-type IL-3 | | | IL-3 K116W | | | P |
|------|-------------------|------|---------|-----|------|---------|-----|-----|
| | | n | $K_D$ (nM) | p | n | $K_D$ (nM) | p | |
| CTLL-2 | IL3Rα + βc | 2 | 0.34 ± 0.03 | — | 2 | 0.14 ± 0.01 | — | — |
| TF-1 | Endogenous | 4 | 2.06 ± 0.35 | — | 3 | 0.35 ± 0.12 | — | 0.006 |
| TF-1Hi | Overexpressed IL3Rα | 7 | 0.14 ± 0.02 | <0.0001 | 6 | 0.065 ± 0.006 | 0.003 | 0.004 |
| COS | IL3Rα + βc | 15 | 0.48 ± 0.05 | — | 6 | 0.43 ± 0.09 | — | 0.98 |
| COS | IL3Rα SP2 + βc | 10 | 7.61 ± 0.97 | <0.0001 | 6 | 2.98 ± 0.87 | 0.011 | 0.004 |
| COS | IL3Rα C76A,C195A + βc | 8 | 1.10 ± 0.31 | 0.009 | 6 | 0.94 ± 0.26 | 0.11 | 0.69 |
| COS | IL3Rα D197L + βc | 6 | 4.37 ± 0.85 | <0.0001 | 4 | 1.60 ± 0.09 | <0.0001 | 0.02 |
| COS | IL3Rα | 8 | 140.6 ± 16.9 | — | 6 | 17.8 ± 5.8 | — | <0.0001 |
| COS | IL3Rα SP2 | 3 | N.B. | — | 6 | N.B. | — | — |
| COS | IL3Rα C76A,C195A | | n.d. | — | 4 | N.B. | — | — |
| COS | IL3Rα D197L | | n.d. | — | 4 | N.B. | — | — |

Binding of wild-type IL-3 or IL-3 K116W to cells expressing the indicated IL-3 receptor subunits was measured in saturation binding assays using radioiodinated cytokine[60, 66]. The CTLL-2 cell line expresses full-length IL3Rα and βc was used as a basis for functional studies of IL3Rα mutants. The human TF-1 cell line or its derivative, TF-1 Hi that expresses high levels of IL3Rα as a result of being transduced with a lentivirus encoding human IL3Rα, were used as a model of human AML. COS cells were transfected with plasmids encoding wild-type, D197L or C76A,C195A forms of full-length IL3Rα or IL3Rα SP2, alone or together with βc. Errors represent SEM from the indicated number (n) of experiments or SD for n < 3. Statistical significance of differences in binding between TF-1 and TF-1Hi or wild-type IL3Rα and the mutant forms of IL3Rα (p) or between IL-3 and IL-3 K116W (P) was determined using a two-tailed unpaired t test. Related to Figs. 3, 4, 5, 6, Supplementary Figs. 4, 6, 8 and Table 3
n.d. not done, N.B. no binding

interaction, such as V201, K235 or E276, preferentially reduced wild-type IL-3 function (Supplementary Table 2) and suggests that these residues indirectly influence the K116/W116 cytokine interaction. We were surprised that many IL3Rα NTD mutations (ie, site 1b) also preferentially reduced wild-type IL-3 function, since the IL3Rα NTD residues are distal to site 1a and the interaction with IL-3 K116/W116 (Figs. 1, 3, Supplementary Table 2). These results suggest that the IL3Rα NTD plays a more dynamic role in IL-3 recognition and function than was previously anticipated.

We were also surprised that wild-type IL-3 and IL-3 K116W were indistinguishable at stimulating the proliferation of CTLL-2 cells expressing IL3Rα and βc (ED$_{50}$ = 0.028 ng ml$^{-1}$ and 0.018 ng ml$^{-1}$ respectively, Supplementary Table 2). Similarly, we observed that the binding affinity of wild-type IL-3 and IL-3 K116W to CTLL-2 cells expressing IL3Rα and βc was indistinguishable ($K_D$ = 0.34 nM vs 0.14 nM respectively, Table 2). Comparable binding affinities were also obtained using COS cells transiently expressing IL3Rα and βc ($K_D$ = 0.48 nM vs 0.43 nM respectively, Table 2). In contrast, using the TF-1 human erythroleukaemic cell line, IL-3 K116W displayed the expected superkine behaviour with enhanced binding ($K_D$ = 0.35 nM vs 2.06 nM respectively, Table 2) and function (ED$_{50}$ = 0.008 ng ml$^{-1}$ and 0.044 ng ml$^{-1}$ respectively, Supplementary Fig. 6a) relative to wild-type IL-3. Considering that endogenous IL-3 receptor expression is typically low on primary cells compared to engineered cell lines (Supplementary Fig. 6b), we wondered if this contributed to the differences in IL-3 function. Using a TF-1 cell line engineered to overexpress IL3Rα[26] (Supplementary Fig. 6b), we observed that wild-type IL-3 and IL-3 K116W are functionally indistinguishable (ED$_{50}$ = 0.005 ng ml$^{-1}$ each, Supplementary Fig. 6a), and importantly, while wild-type IL-3 is significantly more potent (approximately ninefold) on cells expressing high levels of IL3Rα, IL-3 K116W is not. Surprisingly, wild-type IL-3 and IL-3 K116W both display comparable and enhanced binding to TF-1Hi cells ($K_D$ = 0.14 nM vs 0.065 nM respectively, Table 2) perhaps reflecting differences in the IL-3 receptor in these cells. Our results indicate that the binding and function of wild-type IL-3 is positively influenced by the level of IL3Rα expression, whereas IL-3 K116W is markedly less influenced in these cells.

**Explanation for the enhanced binding of IL-3 K116W.** The crystal structures of wild-type IL-3 and IL-3 K116W bound to

IL3Rα are very similar (Fig. 4, Supplementary Fig. 2b) and yet the ligand-binding affinity for these two complexes are very different. Compared to wild-type IL-3, IL-3 K116W has enhanced binding to full-length IL3Rα expressed on COS cells ($K_D$ = 140.6 nM vs 17.8 nM, respectively, Table 2) or to soluble IL3Rα ($K_D$ = 510 nM vs 5.3 nM respectively[12]). The increased receptor affinity of IL-3 K116W is due to a combination of an increased association constant and decreased dissociation constant[12]. Examination of the wild-type IL-3 and IL-3 K116W conformational states by MD simulations showed that IL-3 K116W was conformationally restricted compared to wild-type IL-3 (Fig. 5, Supplementary Figs. 4b, c, 7a–c). The reduced conformational flexibility of IL-3 K116W, together with the W116 side chain slotting into a site 1a pocket (Fig. 4b, d), could account for the increased association constant of the IL-3 K116W superkine.

We have previously hypothesised that the IL3Rα NTD is a highly mobile domain as IL3Rα can exist in open and closed conformations based on differences in NTD placement in the crystal structure[11]. IL3Rα transitioning to the closed conformation was proposed to occur upon binding of IL-3 to site 1a and through the resultant engagement of site 1b in the NTD, lead to reduced IL-3 dissociation. The closed IL3Rα conformation is very similar to the conformation observed in the binary complex crystal structures herein (Supplementary Fig. 2b). Interestingly, the smaller root-mean-square fluctuation of the cytokine NTD Cα atoms and the reduced interatomic distance between IL-3 E43–IL3Rα K54 and IL-3 D44–IL3Rα Y58 demonstrates that the IL-3 K116W binary complex has a reduced mobility compared to the wild-type IL-3 binary complex (Fig. 5, Supplementary Fig. 7h, i). When overlaid via the CRM (ie, D2–D3), the conformational stability of the IL3Rα NTD is increased in the wild-type IL-3 binary complex and is most pronounced in the IL-3 K116W binary complex compared to the unliganded (ie, apo) form of IL3Rα (Fig. 5a, b). The reduced mobility of the IL-3 K116W binary complex can also be visualised by the alignment of 10 ns snapshots from the binary complex MD simulations via the cytokine (left hand images in Fig. 5c, d). The conformational variability of IL3Rα over the 100 ns simulation timeframe is far smaller for the IL-3 K116W than the wild-type IL-3 binary complex.

Observing the decreased conformational mobility of IL3Rα NTD in the IL-3 K116W binary complex, we investigated the contribution of the NTD as a whole by examining IL3Rα SP2, a

splice variant of IL3Rα lacking the NTD[24] (Supplementary Fig. [1]c) and with no measurable binding to IL-3 (Table [2]). Emphasising the critical role played by the NTD, we also observed no measurable binding of IL-3 K116W to cells expressing IL3Rα SP2 (Table [2]). To assess the effect of the

NTD upon cytokine:IL3Rα complex stability, the IL3Rα NTD in both binary complexes was truncated and the N-terminal residues mutated in silico to those in the published SP2 sequence (Supplementary Fig. [1]c)[24]. When 10 ns snapshots from the 100 ns binary complex MD simulations were overlaid via the wild-

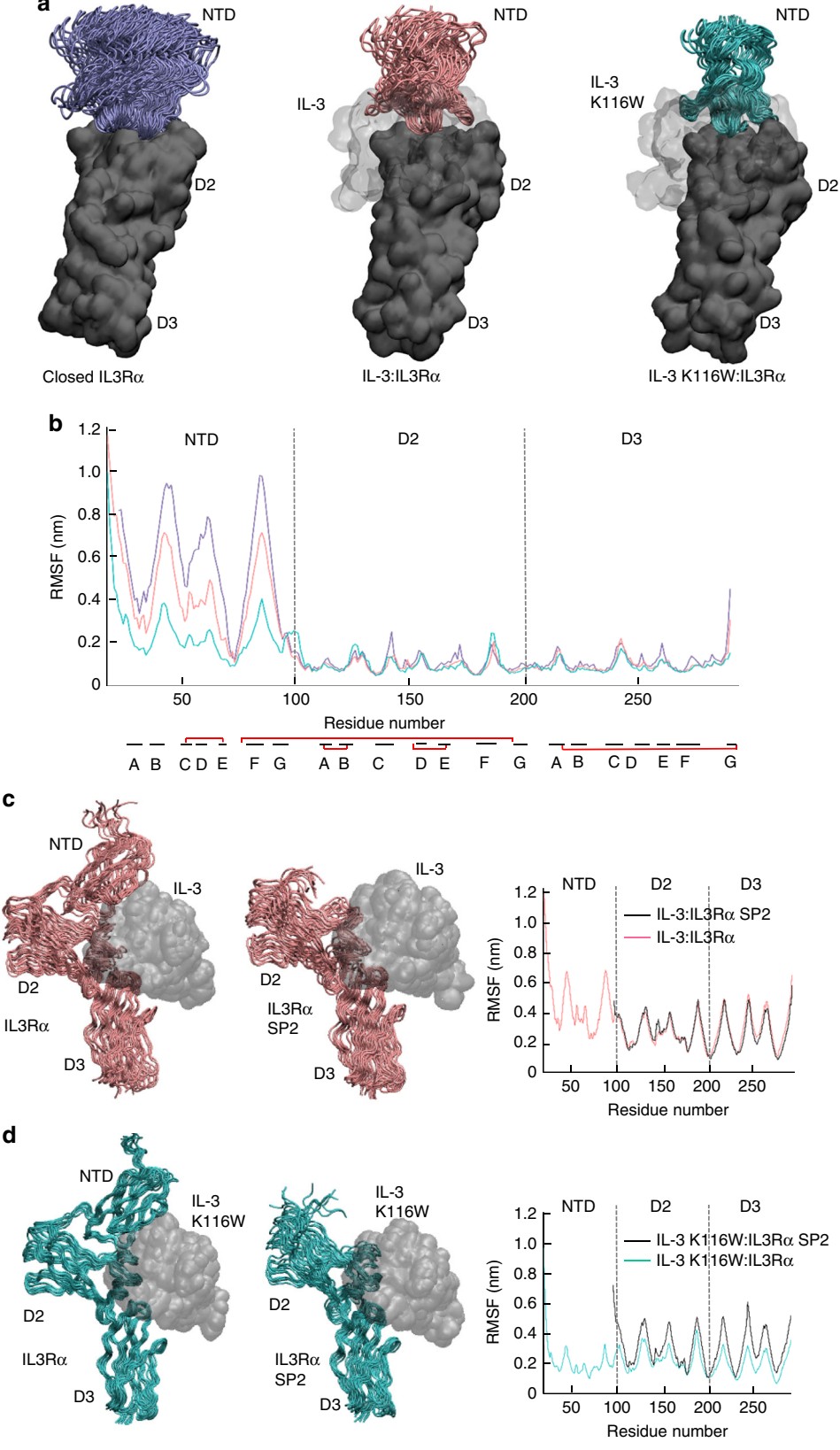

type IL-3, there was little difference in the conformational variability of wild-type or SP2 IL3Rα (Fig. 5c). However, the stabilising role of the NTD is apparent in the simulations of the IL-3 K116W binary complexes (Fig. 5d).

**NTD prevents spontaneous receptor dimerisation.** We next examined what elements in the IL3Rα NTD were important to support its correct conformation to maximise IL-3 binding and signalling. Our previous crystal structure of IL3Rα complexed to an antibody[11] and the structure of IL3Rα complexed to IL-3 presented here (Fig. 1a) indicated that the IL3Rα NTD domain is constrained by a disulphide bond between C76 in the NTD and C195 in D2, and by a hydrogen bond between S74 in the NTD and D197 in D2. We therefore investigated if the functional contribution of the IL3Rα NTD was influenced by mutations that retain the NTD but perturb its presentation to IL-3. The double mutation C76A and C195A abolished the disulphide bond and was therefore predicted to produce IL3Rα with a tethered but highly mobile NTD. Similarly, the IL3Rα mutation D197L was predicted to fill a pocket between the NTD and D2 and, through steric interference, force IL3Rα to adopt an open-like conformation hence also impairing site 1b interactions[11]. MD simulations (75 ns) of the IL3Rα C76A,C195A and IL3Rα D197L mutants suggest an increased mobility for the NTD in both cases (Fig. 6a). Breaking the IL3Rα NTD-D2 inter-domain disulphide bond results in the NTD twisting laterally away from the conformations adopted by the apo wild-type IL3Rα (Fig. 6a, b, d). Increasing the bulk of residue 197 forces the NTD away from D2 and the distance between the S74–L197 Cα atoms for the IL3Rα D197L mutant is similar to that observed for the open apo wild-type IL3Rα (Fig. 6a, c, e). Thus, the IL3Rα C76A,C195A and IL3Rα D197L mutations are both predicted to exhibit 'SP2-like' activities as the NTD is displaced (Fig. 6d, e) and no longer plays a stabilising role in cytokine binding.

We were unable to detect direct binding of IL-3 K116W to COS cells expressing the IL3Rα C76A,C195A or IL3Rα D197L mutants, which was in contrast to the relatively high-affinity binding we observed on cells expressing full-length IL3Rα but comparable to the lack of direct binding we observed for the IL3Rα SP2 isoform (Table 2). When co-expressed with βc, IL3Rα SP2 and IL3Rα D197L displayed significantly reduced binding affinities for wild-type IL-3 and IL-3 K116W (Table 2). For IL3Rα C76A,C195A, there was a modest but significant reduction in binding affinity for wild-type IL-3 but not IL-3 K116W (Table 2).

To examine the functional capacity of the constrained NTD mutants, we examined the response of mouse FDH cells expressing these receptors. Cells co-expressing βc and IL3Rα SP2, IL3Rα C76A,C195A and IL3Rα D197L proliferated in response to wild-type IL-3 and IL-3 K116W stimulation but with reduced potency compared to cells expressing full-length IL3Rα

($ED_{50} = 0.29$ ng ml$^{-1}$ and 0.37 ng ml$^{-1}$ respectively, Table 3). Cells expressing IL3Rα SP2 and IL3Rα C76A,C195A had comparable functional responses to wild-type IL-3 ($ED_{50} = 6.94$ ng ml$^{-1}$ and 12.17 ng ml$^{-1}$ respectively, Table 3) and IL-3 K116W ($ED_{50} = 2.19$ ng ml$^{-1}$ and 2.51 ng ml$^{-1}$ respectively, Table 3). In contrast, cells expressing IL3Rα D197L displayed a modest reduction in cell proliferation to both wild-type IL-3 and IL-3 K116W ($ED_{50} = 1.71$ ng ml$^{-1}$ and 0.60 ng ml$^{-1}$ respectively, Table 3). Therefore, elimination of the C76–C195 disulphide bond that constrains NTD mobility or substitutions at D197 that prevent proper engagement of the NTD, produce full-length IL3Rα that has functional similarities with IL3Rα SP2. Modest improvements in binding affinity (Table 2) and potency (Table 3) for IL-3 K116W relative to wild-type IL-3 are consistent with IL-3 K116W having an enhanced interaction with IL3Rα D2–D3. Collectively, these results demonstrate that the NTD interaction with IL-3 is exquisitely sensitive to conformational changes.

We next examined whether the IL3Rα NTD was important for receptor heterodimerisation and tyrosine phosphorylation as a measure of receptor signalling. Using HEK293-T cells stably expressing βc and transiently co-expressing IL3Rα SP2, IL3Rα C76A,C195A and IL3Rα D197L (Supplementary Fig. 8), we observed IL-3-dependent, co-immunoprecipitation (IP) of Flag-tagged IL3Rα following βc IP that was associated with tyrosine phosphorylation of βc (Fig. 6f). Surprisingly, deletion of the NTD was associated with constitutive association of IL3Rα SP2 with βc but the retention of IL-3-dependent tyrosine phosphorylation of βc (Fig. 6f). Consistent with their SP2-like properties, IL3Rα C76A,C195A, and to a lesser extent, IL3Rα D197L displayed constitutive association with βc but retained IL-3-dependent tyrosine phosphorylation of βc (Fig. 6f). These data strongly suggest that a naturally constrained IL3Rα NTD prevents spontaneous IL3Rα heterodimerisation with βc and could potentially protect cells from unregulated signalling.

**Discussion**

We report here the three-dimensional atomic structures of the IL-3 receptor α-subunit in complex with IL-3 and with the IL-3 K116W superkine, revealing the key interactions between receptor and cytokine that compose site 1, the first and major recognition step that initiates signalling in this family of receptors. Of particular significance, is the finding through extensive analyses by functional and MD simulation approaches that the NTD of IL3Rα exhibits exquisite flexibility and a dynamic transition from a high to a low mobility state upon cytokine binding that leads to optimal ligand recognition, receptor dimerisation and signalling. Since the NTD is also a feature of other members of the type I cytokine receptor family, the results are likely to have general implications for our understanding of how the NTDs function in receptor assembly and signalling in other cytokine receptors.

---

**Fig. 5** Ensemble of IL3Rα MD simulations in isolation and in complex with wild-type IL-3 and IL-3 K116W. **a** Snapshots from 75 ns (unliganded/apo IL3Rα) and 100 ns (binary complexes) MD simulations overlaid via the IL3Rα CRM (ie, D2–D3). The mobility of the NTD progressively decreases when going from the unliganded/apo IL3Rα (purple NTD, left), to the wild-type IL-3 binary complex (pink NTD, middle) and then to the IL-3 K116W binary complex (green NTD, right). The IL3Rα CRM is shown as a dark grey molecular surface, the NTD is shown as a cartoon and the cytokine is depicted as a light grey molecular surface. **b** Root-mean-square fluctuation (RMSF) of the Cα atoms in IL3Rα provides a measure of atom mobility around its average position during the simulation. The data demonstrates that IL-3 K116W stabilises the NTD in the binary complex. Colour scheme and IL3Rα alignment as for **a**. The location of the β-strands (black lines) and the disulphide bonds (red lines) are shown below the X-axis. **c** Comparison of full-length IL3Rα vs IL3Rα SP2 mobility in the wild-type IL-3 binary complex. Snapshots from the 100 ns MD simulations overlaid via the cytokine (grey molecular surface). IL3Rα shown as a pink cartoon. Loss of the NTD has little impact upon the conformational mobility of the α-subunit. **d** Comparison of full-length IL3Rα vs IL3Rα SP2 mobility in the IL-3 K116W binary complex. Snapshots from the 100 ns MD simulations overlaid via the cytokine (grey molecular surface). Il3Rα shown as a green cartoon. Loss of the NTD has a visible negative impact upon the conformational mobility of the α-subunit and it now resembles the wild-type IL-3 system shown in **c**

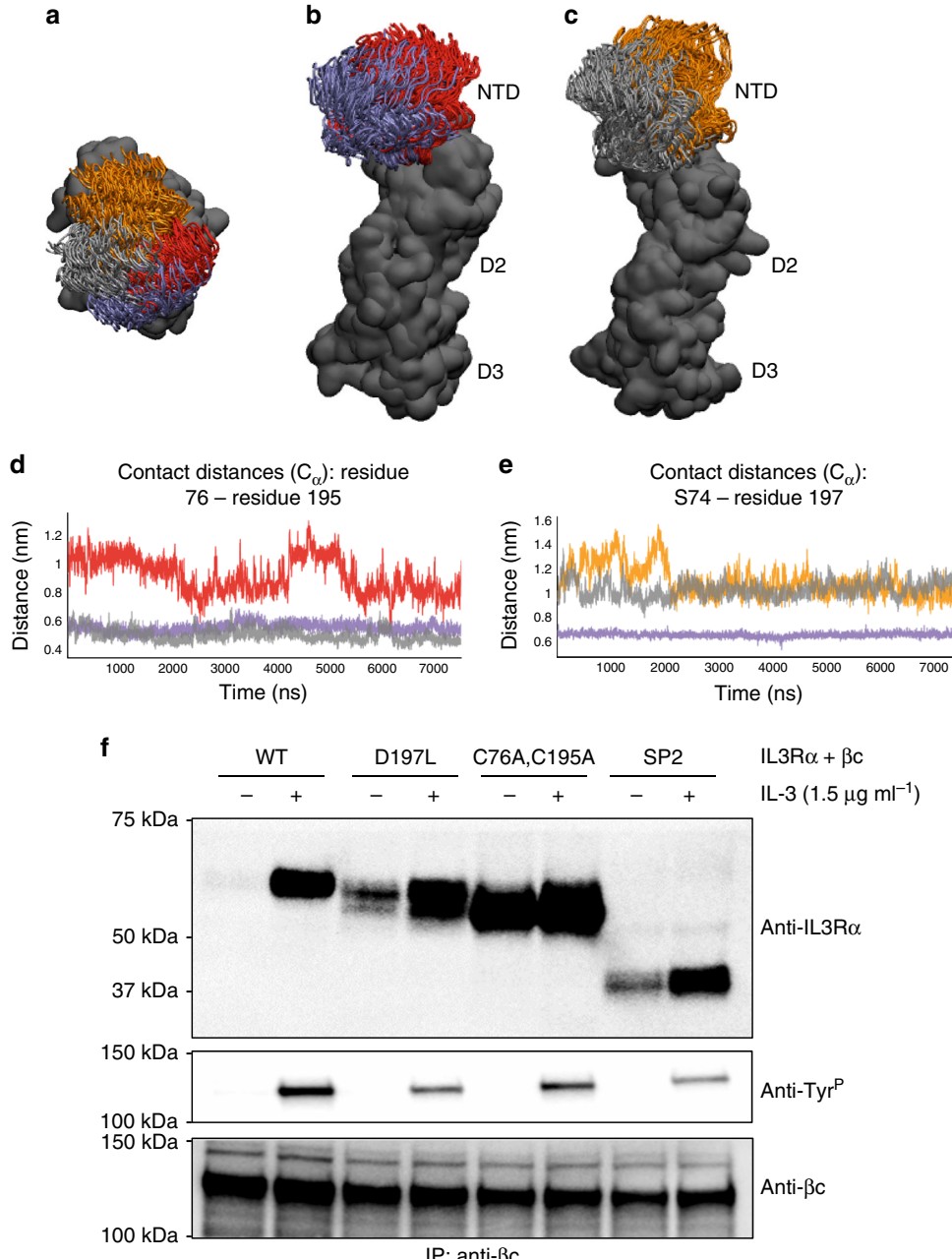

**Fig. 6** IL3Rα mutations that perturb NTD placement also promote spontaneous receptor heterodimerisation. Snapshots from apo IL3Rα MD simulations (75 ns wild-type, 100 ns mutants) overlaid via the IL3Rα CRM. Simulations initiated from both the closed and open wild-type apo IL3Rα conformations. **a** When looking down onto the NTD, the range of conformations sampled by the apo wild-type IL3Rα closed (purple) and open (grey) starting conformations, the C76A,C195A mutant (red) and D197L mutant (orange) can be clearly observed. **b** In the IL3Rα C76A,C195A mutant, the NTD twists away from those conformations adopted by the closed (or open, not shown) apo wild-type IL3Rα. **c** In the IL3Rα D197L mutant, the NTD is also forced away from D2 when compared to the conformations adopted by the open (or closed, not shown) apo wild-type IL3Rα. **a–c** the IL3Rα CRM is depicted as a dark grey molecular surface and the NTD as a cartoon. View in **b** and **c** is a rotation of +90° about the X-axis from that shown in **a**. **d** Distance between the Cα atoms of the C76–C195 inter-domain disulphide bond in the closed (purple) and open (grey) starting IL3Rα conformations compared to that in the C76A, C195A mutant (red). The greater distance between the NTD and D2 for the C76A,C195A mutant is consistent with the reduced binding affinity for IL-3. **e** Distance between the Cα atoms of S74-D197 in the closed (purple) and open (grey) starting conformations compared to that in the IL3Rα D197L mutant (orange). In the closed wild-type IL3Rα conformation, S74 (NTD) and D197 (D2) form an inter-domain hydrogen bond, whereas the distance is too great in the open conformation. The leucine side chain of the D197L mutant keeps the NTD and D2 apart in an open-like conformation. **f** HEK293T cells stably expressing βc were transfected with either wild-type (WT), D197L, C76A,C195A or SP2 forms of Flag-tagged IL3Rα and stimulated with 1.5 µg ml⁻¹ IL-3 for 45 min at 4 °C. Cell lysates were immunoprecipitated with the βc antibody BION-1. Immunoprecipitates were immunoblotted with antibodies for IL3Rα (anti-IL3Rα), phosphotyrosine (anti-Tyr^P) or total βc (anti-βc)

**Table 3 Enhanced site 1a interactions do not fully compensate for perturbations of functional interactions through the NTD**

| IL3Rα | IL-3 | | | IL-3 K116W | | | P |
|---|---|---|---|---|---|---|---|
| | n | ED$_{50}$ (ng ml$^{-1}$) | p | n | ED$_{50}$ (ng ml$^{-1}$) | p | |
| Wild type | 6 | 0.29 ± 0.11 | — | 6 | 0.37 ± 0.10 | — | 0.58 |
| SP2 | 3 | 6.94 ± 1.19 | 0.0003 | 3 | 2.19 ± 0.15 | <0.0001 | 0.02 |
| C76A,C195A | 3 | 12.17 ± 2.02 | 0.0002 | 3 | 2.51 ± 0.49 | 0.002 | 0.01 |
| D197L | 3 | 1.71 ± 0.08 | 0.0002 | 3 | 0.60 ± 0.13 | 0.27 | 0.002 |

Functional activity of IL3Rα mutants was measured in IL-3 or IL-3 K116W dose–response proliferation assays using FDH cell lines transduced to co-express the βc subunit with either wild-type or mutant forms of IL3Rα. Half-maximal responses (ED$_{50}$) were calculated from each experiment and averaged. Errors represent SEM from the indicated number (n) of experiments. Statistical significance of functional response between wild-type IL3Rα and the mutant forms of IL3Rα (p) or between IL-3 and IL-3 K116W (P) was determined using a two-tailed paired t test. Related to Figs. 5, 6, Supplementary Fig. 8 and Table 2

A comparison of the IL-3 binary complexes showed that the IL3Rα site 1a interaction involves the elbow region of D2 and D3 (Fig. 1a, c) that follows the paradigm established by the growth hormone receptor[25]. In both complexes, a large interface was seen with mutations of Q178, V201, Q204 and E276 in IL3Rα having a significant effect upon wild-type IL-3 and IL-3 K116W signalling (Supplementary Table 2). In contrast, mutations of N233 and Y279 in IL3Rα had a significant effect upon wild-type IL-3 but not IL-3 K116W signalling indicating an altered role for interactions through these residues for IL-3 K116W. The major difference observed between the two complexes was the side chain of the superkine's W116 that inserts itself in a groove formed by Q204 and Y279 of IL3Rα (Fig. 4), with mutation of Q204 reducing both wild-type IL-3 and IL-3 K116W signalling (Supplementary Table 2).

Both IL-3 binary complexes have a very similar site 1b interface, yet they display important functional differences. The D-strand of the IL3Rα NTD shows a close interaction with the AB loop and CD loop of IL-3 (Fig. 1b). This helps to explain why mutations of IL-3 E43 or D44 in the α2-helix of the AB loop disrupt IL-3 function[16], whereas mutation of adjacent residues in the α2-helix including G42, Q45 and D46 enhances IL-3 binding to IL3Rα as well as IL-3 function[12,16]. Furthermore, mutagenesis of IL3Rα NTD residues K54, Y58 or G71 abolishes direct IL-3 binding[11]. The IL3Rα NTD EF loop is stabilised by an interaction with the NTD B-strand residue Q37, which makes main-chain contacts through its side chain with I73, and carbonyl interactions with Q69 and F70. Mutation of IL3Rα Q37 also reduces IL-3 binding[11], presumably as a result of changes to the NTD EF loop conformation. Despite marked differences in the specific interactions that comprise site 1b, the surface area between cytokine and IL3Rα at the site 1b interface is relatively similar among the related receptor complexes (34% in IL3Rα, 42% in GMRα (PDB ID: 4RS1[11]), 64% in IL5Rα (PDB ID: 3VA2[21]), 47% in IL13Rα1 (PDB ID: 3BPO[22]) and 34% in IL13Rα2 (PDB ID: 3LB6[23])) (Fig. 2, Supplementary Fig. 2; see additional detail in Supplementary Note 1) suggesting that it may be serving similar functions in these other receptors. Interestingly, despite there being essentially no difference in the way that IL-3 and IL-3 K116W interacted with IL3Rα through site 1b, there were some crucial differences in how mutation of many of these residues selectively impaired the wild-type IL-3 function. Unlike IL3Rα interactions with IL-3, the interaction with IL-3 K116W was not markedly affected by mutations of K54, Y58 or G71 in the NTD (Fig. 3, Supplementary Table 2) indicating that the K116W mutation has stabilised the interaction with IL3Rα through distal effects on the NTD.

In the published structure of IL3Rα bound to the blocking antibody CSL362, we found the NTD in two distinct structural forms, an open and a closed conformation[11] that led us to hypothesise that the IL3Rα NTD may be a highly mobile domain. We now show by MD simulations that indeed the NTD is highly mobile and that the presence of IL-3 stabilises the NTD in the binary complex. The largest effect was observed for IL-3 K116W, which restricts the conformational flexibility of the binary complex (Figs. 4, 5a, b), probably as a result of W116 inducing a distal effect that stabilises the engagement with the NTD thus restricting domain mobility. This high mobility may be a general feature of the NTD of the type I cytokine receptor family and a major reason for the difficulty in their crystallisation, particularly in the ligand-free state. The flexibility and high mobility of the NTD may facilitate locking of the cytokine upon binding to site 1a and the appropriate presentation of the cytokine:α-subunit complex to the signalling subunit. Similarly, in the interferon (IFN) receptor, IFNAR1, conformational flexibility has been observed in the IFN-bound and unliganded/apo structures. Comparison of apo IFNAR1 with its IFN-bound form revealed a large spatial rearrangement in the four extracellular subdomains (SD1–SD4)[27]. Although the IFN ligand binds largely to the hinge region between the IFNAR1 SD2 and SD3 domains, the NTD of IFNAR1 (ie, SD1) moves downwards ~10 Å to interact with the IFN ligand, further securing it to IFNAR1. The high mobility of IFNAR1 is required for optimal ligand binding and the formation of a ternary complex that then initiates trans-phosphorylation between JAK1 and TYK2[28].

The importance of a conformationally restricted IL3Rα NTD is further underscored by mutagenesis of the disulphide bond between the NTD and D2 and of the aspartic acid at position 197. These mutations abolished direct binding of IL-3 K116W and reduced high-affinity binding and signalling in the presence of βc (Tables 2, 3). Interestingly, mutations that perturb the position of the NTD, or that completely delete the NTD, are associated with spontaneous heterodimerisation of IL3Rα with βc, although surprisingly, this spontaneous dimerisation does not result in receptor tyrosine phosphorylation (Fig. 6f). These observations are comparable with reports of preformed but non-signalling dimers of receptors for growth hormone[29] and erythropoietin[30] that signal following ligand binding-induced conformational changes in the receptor complex. In an analogous manner, the architecture of the NTD-deficient or NTD-displaced IL3Rα may promote association with βc to form inert IL3Rα:βc complexes that only signal following conformational changes triggered by the binding of IL-3. While tyrosine phosphorylation of the βc receptor was strictly dependent upon IL-3 stimulation, other forms of receptor signalling could be associated with spontaneous heterodimerisation of IL3Rα and βc. Additionally, in cells expressing wild-type IL3Rα, the NTD may serve a protective role by preventing signalling from receptor association arising in the absence of ligand.

The relatively high expression of IL3Rα in a number of leukaemias has presented a window of opportunity to develop new therapeutic approaches directed to IL3Rα such as monoclonal antibodies (MAbs)[8,31,32], T cells engineered to express chimeric antigen receptors (CAR T cells)[33] and toxin conjugates with IL-

3[34–37]. The latter was originally only partially successful as half the samples proved resistant to the fusion protein[38]. The potency of the toxin-IL-3 fusion protein was improved dramatically by introducing the K116W substitution into the IL-3 portion[38–40]. The insights described in our work explain the potency of the IL-3 K116W superkine and suggest new avenues for further protein engineering of even more potent superkines.

In comparing the effects of IL3Rα mutations on IL-3 and IL-3 K116W binding and signalling, we noted that for cells expressing endogenous IL-3 receptors (such as TF-1 cells) there is, as previously reported by us and others, a sixfold increased potency of the IL-3 superkine (Table 2, Supplementary Fig. 6a). However, in cells overexpressing IL3Rα (such as engineered CTLL-2 or FDH cell lines, TF-1Hi cells and COS cells), this difference was diminished or absent (Fig. 3, Supplementary Fig. 6a, Tables 2, 3, Supplementary Table 2). The observation that IL3Rα expression levels vary widely among different primary cell populations[41] suggests the likelihood of differences in IL-3 sensitivity among these cell populations. This may also have implications in AML where the AML leukaemic stem/progenitor cells express elevated levels of IL3Rα[7,8], but may also indicate differences in the composition or form of the endogenous IL-3 receptor. For instance, it is known that transcripts for the IL3Rα SP2 splice variant are expressed by TF-1 cells[24] and this could contribute to reduced IL-3 binding and function compared to cells only expressing full-length IL3Rα.

In summary, our results highlight the unique dynamics and role of the NTD in IL-3 signalling and suggest the purpose of its conservation among members of the type I cytokine receptor family.

## Methods

**Production of recombinant IL-3:sIL3Rα binary complexes**. We prepared wild-type IL-3 and IL-3 K116W binary receptor complexes using partially glycosylated variants of the IL3Rα extracellular domain and truncated wild-type IL-3 or IL-3 K116W. DNA fragments encoding soluble IL3Rα ΔN5 (sIL3Rα; residues L20-S307 of the expressed peptide with the N212Q mutation) or soluble βc ΔN3 (sβc; residues E25-S438 of the expressed peptide with the N346Q mutation) were cloned into the pFastBac1 vector (Invitrogen). The resulting plasmids were transformed into the DH10Bac *Escherichia coli* strain from which recombinant bacmid DNA was isolated and transfected into Sf9 insect cells (ATCC CRL-1711) using Cellfectin reagent (Invitrogen). Following three rounds of recombinant virus propagation, large-scale expression was performed by infection of Sf9 cells grown in Sf-900 II serum-free medium (Life Technologies). The supernatant was collected after 5 days and recombinant proteins affinity-purified using HiTrap NHS-activated HP columns (GE Healthcare) coupled to the IL3Rα-specific MAb 9F5[42] or the βc-specific MAb BION-1[43]. Bound protein was eluted using 500 mM NaCl, 50 mM phosphoric acid pH 2.4 and the eluted protein concentrated and further purified by preparative size exclusion chromatography (SEC) using a Superdex 200 column (26 × 600 mm, GE Healthcare) operated at 2 ml min$^{-1}$ with 150 mM NaCl, 50 mM sodium phosphate pH 7.0 as running buffer[44]. Binding and functional studies with wild-type IL-3 demonstrate that the N212Q mutation has no impact on the function of full-length IL3Rα (Supplementary Fig. 3c, d). Wild-type or K116W versions of human IL-3, comprising residues W13 to Q125 of the mature peptide with a W13Y mutation and a GAMGS N-terminal tail arising from the expression plasmid, were purified from *E. coli*[45]. BL21 CodonPlus (DE3)-RIPL cells transformed with pETNusH expression vectors encoding Nus-6xHis-TEV:IL-3 fusion proteins, were cultured to an OD$_{600}$ of 0.6 and induced with 0.1 mM isopropyl β-D-1-thiogalactopyranoside (IPTG) at 23 °C for 16 h. Cells were lysed and Nus-6xHis-TEV:IL-3 fusion proteins affinity-purified using HisTrap FF columns (GE Healthcare). Fusion proteins were concentrated, dialysed against 50 mM Tris-HCl pH 8.0, adjusted to 1 mM dithiothreitol, 0.5 mM EDTA and incubated with TEV protease at 23 °C for 16 h. IL-3 was purified from the TEV digest by preparative SEC then adjusted to 0.1% (v v$^{-1}$) trifluoroacetic acid, 1% (v v$^{-1}$) acetic acid and further purified by reversed-phase chromatography using an Aquapore RP300 reversed-phase column (4.6 × 100 mm, PerkinElmer) equilibrated in buffer A (0.1% v v$^{-1}$ trifluoroacetic acid) and eluted using a linear gradient from 0 to 100% buffer B (0.085% v v$^{-1}$ trifluoroacetic acid in acetonitrile). Fractions containing purified hIL-3 were lyophilised, resuspended in PBS and sterilised using Corning Spin-X filters (Sigma-Aldrich). We, and others, have previously demonstrated that compared to full-length recombinant IL-3, the truncated IL-3 has equivalent bioactivity[45].

A binary complex consisting of wild-type IL-3 and sIL3Rα ΔN5 was isolated by SEC of a 2:1.5:1 molar ratio mixture of IL-3:sIL3Rα ΔN5:sβc ΔN3 (Supplementary Fig. 1a). Using a similar strategy, a binary complex consisting of IL-3 K116W and

sIL3Rα ΔN5 was isolated by SEC of a 1.6:1 molar ratio mixture of IL-3 K116W: sIL3Rα ΔN5 (Supplementary Fig. 1b).

**Analysis of sIL3Rα N-linked glycosylation mutants**. Soluble IL3Rα complementary DNA (cDNA) with a C-terminal 6xHis tag (sIL3Rα-6xHis) was cloned into the pcDNA3.1 expression vector (Invitrogen) and asparagine to glutamine mutations generated by PCR at the six predicted N-linked glycosylation sites. Freestyle™ 293 suspension cells (Invitrogen) were transiently transfected with the pcDNA:sIL3Rα-6xHis plasmids[11] and the resulting supernatants immunoblotted with 0.9 μg ml$^{-1}$ of the anti-IL3Rα MAb, 7G3[42] and 1 μg ml$^{-1}$ of goat-anti-mouse IgG - fluorescein isothiocyanate (IgG-FITC (Thermofisher 62-6511)) or 0.25 μg ml$^{-1}$ of FITC-conjugated anti-His tag antibody (Anti-6xHis Genescript cat no. A01620).

**Structure determination of IL-3:IL3Rα binary complexes**. Crystallisation was performed using the hanging drop vapour diffusion method at 4 °C. Crystallisation trials of IL-3 K116W:IL3Rα were set up using the protein crystallisation screen JCSG[46] at the CSIRO C3 crystallisation facility (http://crystal.csiro.au) with the protein at a concentration of 12 mg ml$^{-1}$. Protein of 0.15 μl was mixed with 0.15 μl of crystallisation reagent and hung over 50 μl of reservoir solution. Tetragonal crystals appeared after 5 days in 18–23% (w v$^{-1}$) PEG 8000, 200 mM NaCl and 100 mM citrate-phosphate buffer pH range 4.6–5. After screening around this condition, the best crystals were obtained in 20% PEG 8000, 200 mM NaCl and 100 mM citrate-phosphate buffer pH 4.8. The crystals diffracted to a resolution of 2.4 Å in the hexagonal space group P6$_1$. The space group and cell dimensions were consistent with there being one complex in the asymmetric unit giving a Matthews coefficient of 3.0 Da Å$^{-3}$ and an estimated solvent content of 65%. Before data collection the crystals were dipped consecutively into 5% (v v$^{-1}$), 10% (v v$^{-1}$), 15% (v v$^{-1}$) cryomix (8.4% each of ethylene glycol, glycerol, 2-methyl-2,4-pentanediol, sorbitol, PEG 400, and sucrose) in crystallisation buffer.

Data were collected on the MX2 beamline at the Australian Synchrotron (Clayton, Victoria). Data collection was controlled using Blue-Ice software[47] and processed with the HKL2000 suite[48]. The structures were solved with the program Phaser[49] initially trialling a range of search models including IL3Rα open and closed conformations (PDB ID: 4JZJ), IL13Rα1 (PDB ID: 3BPO), IL13Rα2 (PDB ID: 3LB6), IL5Rα (PDB IDs: 3QT2, 3VA2), and the partial GMRα structure from the ternary GM-CSF complex (PDB ID: 3CXE). The final model used for molecular replacement for the IL-3 K116W binary complex was the nuclear magnetic resonance structure for IL-3 (PDB ID: 1JLI) and the IL13Rα2 receptor (PDB ID: 3BL6) as separate search components. The results from Phaser gave a final translation function Z-score of 9.6. One IL-3 K116W:IL3Rα complex was found in the asymmetric unit. Loop regions were manually built in place using COOT[50] and subsequent rounds of refinement using REFMAC[51]. Residues spanning 27–293 of IL3Rα and 12–123 in IL-3 were identified with clear and unambiguous electron density observed at the IL-3:IL3Rα interface. Electron density was not observed for residues 42–50 and 86–90 in IL3Rα, which are located in flexible loop regions. Electron density representing glycan moieties was observed at glycosylation sites N80 and N218 in IL3Rα. The stereochemical quality of the final model correlates well with structures at similar resolutions with only 0.5% of residues in the disallowed regions of the Ramachandran plot. Other stereochemical parameters were are all better than the allowed ranges defined by PROCHECK[52]. Data and refinement statistics are listed in Table 1 and stereo images of portions of the electron density maps in Supplementary Fig. 10. The PISA (Protein Interfaces, Surfaces and Assemblies) server http://www.ebi.ac.uk/msd-srv/prot_int/pistart. html was used for all protein-ligand surface interaction calculations.

We have previously tried exhaustive crystallisation screening and construct optimisation of wild-type IL-3:IL3Rα for many years without success. Our results presented here provide a possible explanation: the mobility of the NTD inhibits crystallisation. Crystals of wild-type IL-3:IL3Rα did not grow under the same condition as IL-3 K116W:IL3Rα; we were only able to crystallise IL-3:IL3Rα when microcrystals of IL-3 K116W:IL3Rα were used to seed crystals of wild-type IL-3: IL3Rα. Drops comprised 0.2 μl of seed solution, 0.4 μl of protein and 0.4 μl of crystallisation reagent and hung over 1 ml of reservoir solution. Tooth-shaped crystals appeared after 5 days in 20–22% PEG 8000, 200 mM NaCl and 100 mM citrate-phosphate buffer pH range 4.8–5. The crystals diffracted to a resolution of 2.7 Å in the hexagonal space group P6$_5$22. The space group and cell dimensions were consistent with there being two complexes in the asymmetric unit giving a Matthews coefficient of 2.8 Da Å$^{-3}$ and an estimated solvent content of 58%. Before data collection, the crystals were cryoprotected as described for the IL-3 K116W: IL3Rα crystals.

The structure of wild-type IL-3:IL3Rα was solved using the refined model of IL-3 K116W:IL3Rα and the programs described above. The results from Phaser gave a final translation function Z-score of 47.1. Two wild-type IL-3:IL3Rα complexes were found in the asymmetric unit. Building and refinement was completed as described above. Residues spanning 25–294 of IL3Rα and 13–121 in wild-type IL-3 were identified with clear and unambiguous electron density observed at the IL-3: IL3Rα interface. Chains G and I were the most complete of the two copies, however electron density was not observed for residues 48 and 143–146 in IL3Rα, which are all located in flexible loop regions. Electron density representing glycan moieties was observed at glycosylation sites N80, N109 and N218 in IL3Rα. The

stereochemical quality of the final model correlates well with structures at similar resolutions with only 0.3% of residues in the disallowed regions of the Ramachandran plot. Other stereochemical parameters were all better than the allowed ranges defined by PROCHECK[52]. Data and refinement statistics are listed in Table 1.

**Molecular dynamics simulations.** Fully atomistic simulations of (a) the apo IL3Rα closed and open conformations (PDB ID: 4JZJ[11]), (b) wild-type IL-3:IL3Rα binary complex (this work), (c) IL-3 K116W:IL3Rα binary complex (this work), (d) wild-type IL-3 extracted from the binary complex crystal structure, (e) IL-3 K116W extracted from the binary complex crystal structure, (f) modelled SP2 versions of the wild-type IL-3 and IL-3 K116W binary complexes, (g) modelled apo IL3Rα D197L and (h) modelled apo IL3Rα C76A,C195A were carried out. Crystallographic waters, sugars and ions and so on were removed from the starting crystal structures. Missing residues or side chains were modelled in using SYBYL-X 2.1.1 (Certara L. P., http://www.tripos.com) or Coot version 0.7[53]. The SP2 version of IL3Rα (Supplementary Fig. 1c)[24] was modelled using SYBYL-X 2.1.1. The D197L and C76A,C195A IL3Rα mutations were made in silico using SYBYL-X 2.1.1. Hydrogens were added and the protein termini capped with neutral acid and amide groups. Titratable residues were left in their dominant protonation state at pH 7.0. All simulations were performed using the program NAMD-2.10[54] with the CHARMM36[55] force field at a constant temperature of 310 K and atmospheric pressure. Ions were added to balance charge and give an ionic concentration of 150 mM NaCl. The TIP3P water model[56] was used in rhombic dodecahedral simulation cells with at least 12-Å separation between the protein and its periodic image. For long range interactions, the Particle Mesh Ewald method was used with a switching distance of 10 Å and cutoff distance of 12 Å. Simulations were typically performed as follows: the energy was minimised for 5000 iterations, followed by system equilibration for 1 ns and then production simulations of 75–200 ns were carried out. The positions of all atoms in the system were saved every 10 ps for later analysis and visualisation using the programs mdtraj[57] and VMD[58].

**Functional studies with IL3Rα mutants.** Human IL3Rα cDNA was cloned into the pcDNA3.1 (Invitrogen) or pSG5 (Stratagene) expression vectors and mutations generated by PCR[11]. For site 1 functional studies (Fig. 3, Supplementary Table 2) and studies of the NTD interaction (Fig. 6, Table 3), we used an IL3Rα cDNA that encodes a splice variant which lacks three nucleotides and changes N144 and R145 to K144 compared to the sequence of the sIL3Rα used for the structural studies (Supplementary Fig. 1c). The two variants are indistinguishable in IL-3 binding and functional studies[11]. For the spontaneous heterodimerisation studies (Fig. 6, Supplementary Fig. 8), we used an IL3Rα cDNA engineered to include a C-terminal Flag epitope tag (…VQKTALDYKDDDDKA-COOH) following the final threonine residue of full-length IL3Rα. Human βc cDNA was cloned in the pSG5 vector.

Wild-type and mutant IL3Rα was subcloned into the retroviral expression vector pRufHygro and the pRufHygro:IL3Rα plasmids co-transfected into HEK293-T (ATCC CRL-3216) cells with the pEQ-Eco packaging plasmid[59] using Lipofectamine 2000 (Invitrogen) to generate recombinant retrovirus. Wild-type βc cDNA was cloned in the pRufPuro retroviral expression vector and the pRufPuro:βc plasmid used to generate recombinant retrovirus by the same approach. Retrovirus was collected from conditioned medium at day 3 post transfection by centrifugation at 1500 r.p.m. for 5 min and the supernatant 0.45 μm filtered. The CTLL-2 cell line (ATCC TIB-214) is maintained in RPMI media (Gibco) supplemented with antibiotics, 10% (v v⁻¹) foetal bovine serum, 1 mM sodium pyruvate (Gibco), 55 μM 2-mercaptoethanol (Gibco) and 400 U ml⁻¹ murine IL-2. CTLL-2 cells lines were made by transduction with retrovirus. Non-tissue culture treated plates (Falcon) were coated with 32 μg ml⁻¹ RetroNectin (Takara) overnight at 4 °C, blocked at room temperature with 2% w v⁻¹ BSA for 1 h and then centrifuged with retrovirus at 3500 r.p.m. for 1 h at 37 °C. The virus coated plates were then incubated with 1 × 10⁵ CTLL-2 cells in 1 ml complete media overnight at 37 °C and a second round of transduction was performed the following day using fresh retrovirus. Media was changed 24 h following the second round of transduction, and 48 h following this, antibiotic selection was added as described below. Pools of resistant cells grew out over 7–10 days and cells were sorted for receptor expression by flow cytometry. CTLL-2 cells were initially transduced to express βc, selected with puromycin (selection 2 μg ml⁻¹, maintenance 1 μg ml⁻¹) and sorted for βc expression by flow cytometry[60]. Subsequently the CTLL-2:βc cells were transduced to express the IL3Rα variants, selected with hygromycin (selection 1000 μg ml⁻¹, maintenance 500 μg ml⁻¹) and sorted for IL3Rα expression by flow cytometry[60].

The HoxA9-immortalised cell line, FDH, was generated by transduction of foetal liver cells from mice lacking βc and β_{IL-3}[61] using a pFTREtight MCS rtTAadvanced GFP lentivirus encoding a doxycycline-inducible HoxA9-Flag expression cassette[62]. The FDH cell line is maintained in IMDM media (Gibco) supplemented with antibiotics, 10% (v v⁻¹) foetal bovine serum, 100 ng ml⁻¹ murine stem cell factor and 0.5 μg ml⁻¹ doxycycline hyclate (Sigma). FDH cells were transduced with βc, selected with puromycin (selection 2 μg ml⁻¹, maintenance 1 μg ml⁻¹), and subsequently transduced to express the IL3Rα variants and selected with hygromycin (selection 500 μg ml⁻¹, maintenance 250 μg ml⁻¹), with sorting as described above.

Cell proliferation was assessed using CellTiter 96® AQueous (Promega) following the manufacturer's protocol. Data were analysed using GraphPad Prism to determine ED_{50} values.

**Statistical methods.** A two-tailed unpaired t test was used for calculation of significance in binding and functional studies. For figures, error bars are SEM.

**Cell-surface IL-3-binding assays.** COS-7 cells (ATCC CRL-1651) were electroporated with pcDNA3.1:IL3Rα plasmids encoding wild-type or mutant IL3Rα, either alone or in the presence of the pSG5H:βc plasmid encoding human βc[60]. Cell-surface expression of receptor subunits was confirmed by flow cytometry. Wild-type IL-3 and IL-3 K116W were radioiodinated with ¹²⁵I (PerkinElmer) using Pierce pre-coated iodination tubes (Thermo Scientific). Saturation binding assays were performed on transfected COS cells, CTLL-2 cell lines, TF-1 cells or TF-1Hi cells using radioiodinated IL-3[60]. Dissociation constants were calculated using the EBDA and LIGAND programs[63] (KELL Radlig, Biosoft, UK). Representative data from IL-3-binding assays on TF-1 and TF-1Hi cells lines were also analysed by nonlinear regression using GraphPad Prism and yielded essentially identical dissociation constants.

**Assessing the cell-surface expression of IL-3 receptor subunits.** Cell-surface expression of receptor subunits was assessed by flow cytometry using 10 μg ml⁻¹ of MAb's specific for IL3Rα, including 7G3, 9F5[42] and MAB301 (R&D Systems cat no. MAB301), and for βc, 1C1[64]. Bound antibodies were stained using 5 μg ml⁻¹ of PE conjugated goat-anti-mouse IgG (Southern Biotech cat. no. 1030-09). MAb's 7G3 and 9F5 bind IL3Rα NTD and MAB301 binds IL3Rα D2–D3. An irrelevant mouse antibody 1B5 (anti-Giardia IgG₁) was used as a negative control.

**IL3Rα expression and IL-3 function on TF-1 and TF-1Hi cells.** The human erythroleukaemia cell line, TF-1[65] (ATCC CRL-2003) was transduced with lentiviral vectors encoding GFP alone or IL3Rα and GFP[26] to produce the TF-1EV and TF-1Hi lines, respectively. Cell-surface expression levels of IL3Rα and βc was measured by flow cytometry. Wild-type IL-3- and IL-3 K116W-mediated proliferation of TF-1 EV and TF-1Hi lines was assessed using CellTiter 96® AQ_{ueous} (Promega). Data were analysed using GraphPad Prism to determine ED_{50} values.

**Constitutive association of IL3Rα NTD mutants with βc.** HEK293-T cells were stably transduced with pRufPuro:βc to express wild-type human βc as described above. The HEK293-T-βc cells were transfected with 2–10 μg of pcDNA3.1:IL3Rα Flag plasmids encoding wild-type or mutant IL3Rα using Lipofectamine 2000 (Invitrogen). Cells were collected after 2 days and stimulated with 1.5 μg ml⁻¹ IL-3 for 45 min on ice prior to lysis in NP-40 lysis buffer (1% NP-40 (v v⁻¹), 150 mM NaCl, 50 mM Tris-HCl pH 8.0 and 10% (v v⁻¹) glycerol) for 20 min on ice. The βc complexes were immunoprecipitated from clarified lysate using 1C1-conjugated Sepharose beads, washed three times with lysis buffer, fractionated by SDS-PAGE and immunoblotted with antibodies for IL3Rα-Flag (1 μg ml⁻¹ anti-Flag M2:HRP, Sigma cat. no. A8592), βc (1.8 μg ml⁻¹ BION-1[43]) and 0.1 μg ml⁻¹ goat-anti-mouse: HRP (Pierce cat. no. 31437) or phosphorylated tyrosine residues (0.25 μg ml⁻¹ anti-phosphotyrosine:HRP, BD Transduction Laboratories cat. no. 610012). (Uncropped scan of immunoblot Supplementary Fig. 9). Cell-surface expression of receptor subunits was confirmed by flow cytometry of collected cells as described above.

**Data availability.** Atomic coordinates and structure factors of the wild-type IL-3 and IL-3 K116W binary receptor complexes have been deposited in the Protein Data Bank under the ID codes 5UV8 and 5UWC, respectively. All other data are available from the corresponding authors upon reasonable request.

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

## Acknowledgements

This research was partly undertaken on the MX2 beamline at the Australian Synchrotron, Victoria, Australia and we thank the beamline staff for their assistance. We thank Anna Sapa for technical assistance, Denis Tvorogov for the HEK293-T cells expressing βc, Hayley Ramshaw for the TF-1Hi cell line, Paul Ekert (Murdoch Children's Research Institute) for the HoxA9 expression construct, Tony Cambareri for the 1B5 hybridoma and Joanna Woodcock for helpful discussions. We acknowledge the use of the CSIRO Collaborative Crystallisation Centre (C3), Melbourne, Australia for our initial crystallisation studies, and the SA Pathology Detmold Family Cytometry Centre for use of flow cytometry facilities. This research was supported by a Victorian Life Sciences Computation Initiative (VLSCI) grant number RA0002 on its Peak Computing Facility at the University of Melbourne, an initiative of the Victorian Government, Australia. This work was supported by grants from the National Health and Medical Research Council of Australia (NHMRC) to T.R.H., U.D., M.W.P. and A.F.L., Cure Cancer Australia to S.E.B., Cancer Council SA Beat Cancer Fund to T.P.H. and from the Australian Cancer Research Foundation to M.W.P. Funding from the Victorian Government Operational Infrastructure Support Scheme to St Vincent's Institute is acknowledged. S.E.B is a Postdoctoral Fellow supported by the Leukaemia Foundation. T.P.H. is an NHMRC Practitioner Fellow and M.W.P. is an NHMRC Research Fellow.

## Author contributions

S.E.B. performed the crystallographic studies and the structural analyses. T.R.H. prepared proteins and complexes for structural and functional studies. T.R.H., W.L.K., M.D., C.J. M. M.P.H., N.J.W. and E.F.B. did binding and functional studies. T.L.N. did molecular modelling. U.D., C.J.M. and K.S.C.T.S. assisted with structural analyses. T.R.H., M.W.P. and A.F.L. designed experiments. M.T.D. and C.S. performed the MD simulations and analyses. K.S.C.T.S., S.E.B., T.R.H., C.S. and T.L.N. prepared figures. S.E.B., T.R. H., T.P.H., T.L.N., M.W.P., and A.F.L. wrote the paper. All authors discussed the results and commented on the manuscript. M.W.P. and A.F.L. supervised the research.

## Additional information

**Competing interests:** M.P.H. and N.J.W. are employees of CSL Limited and M.T.D. and C.S. are employees of IBM Research Australia. The remaining authors declare no competing financial interests.

