## [Peer Review File · Nature Communications]

Reviewers' comments:

Reviewer #1 (Remarks to the Author):

This is an interesting and technically strong manuscript describing the generation of high affinity IL-3 variants and subsequently determining the crystal structure of a complex between one "superkine" in particular, in complex with the IL-3R-alpha. The authors convincingly show that the D1 (or NTD) of IL-3Ra is highly mobile in the unliganded and wt IL-3 bound states, but that the high affinity cytokine significantly reduces its conformational mobility. Therefore the mechanism of the affinity increase appears to be through a rather novel reduction in the conformational entropy of the receptor. This would be useful for the development of enhanced IL-3 based therapeutic cytokines. I have no significant qualms with publication of this work, although it would help to have thermodynamic measurements of the wt versus superkine binding to IL-3Ra that give some hint about the energetics of the respective interactions, though I do not require this for publication.

The manuscript would be strengthened by discussion of a related phenomenon seen in the IFNAR1 D1 domain that appears to be 'locked down' upon IFN binding, see works from Schreiber and Garcia on this.

minor comment: it is surprising that the authors use the term "superkine" without referencing the original use of the term from Levin et al., Nature 2012.

Reviewer #2 (Remarks to the Author):

In this manuscript, the authors combine crystallography and simulations to address the binding properties of IL-3. With regards to simulations, the authors apply molecular dynamics simulations using an all-atom explicit-solvent model. The methods indicate that the simulations were performed for 50-200ns, each. Overall, the performed simulations are technically sound. However, there are some important questions with regards to data analysis, and changes to the text are necessary.

1) In the main text, the results are simply referred to "MD simulations". Since molecular dynamics is a method, not a model, the term "MD simulations" is quite vague. For example, one could perform MD simulations using a coarse-grained model, rather than an all-atom model. At first usage, the authors should indicate that these are MD simulations with an all-atom explicit-solvent model.

2) The manuscript misuses the term "stability". That is, the section repeatedly describes the differences in the simulations as indicators of stability, though free-energy differences are not being evaluated. The authors should use more descriptive terms that reflect the quantities that are being calculated. It appears that the only analysis provided is time traces of interatomic distances. The claims of the main text (as written) are very broad (e.g. changes in stability) and are completely unsupported by the accompanying figures. It would be helpful if the claims are more precisely stated.

3) While the methods state that simulations were performed for 50-200ns, figure S4 shows a 10ns trajectory and the main text interprets the time trace as indicating changes in confinement. Accordingly, it is unclear whether the simulations were actually 50-200ns, or not. A more general issue with this description is that the analysis does not appear to be systematic. For example, figure S4 is supposed to show that one system is more conformationally restricted, but the simulation appears to be only a few nanoseconds. Inspection of the traces in S4 show that both exhibit large fluctuations. Thus, it is not clear that the differences are beyond the uncertainty due to sampling limitations.

Reviewer #3 (Remarks to the Author):

The manuscript reports two new X-ray structures for IL-3 receptor alpha complexed with IL-3. One is with IL-3 and the other is with the IL-3 mutant K116W. This group has previously reported the structure of IL-3 receptor alpha complexed with a Fab antibody fragment. The structures provide new information for understanding the molecular details of the interaction of IL-3 and IL-3K116W with IL-3 receptor alpha. This receptor is important in haematopoiesis and appears to play a role in AML so understanding its function is significant.

For crystallization, an N212Q mutant of the receptor was used. It is important to state if this mutation affects binding or signalling. Fig1b (suppl) shows significant levels of impurities in the K116W complex. It is essential to verify these are absent from the crystals.

The sequence of the alpha receptor is given. What sequence was used for the SP2 isoform?

Data availability: The atomic coordinates and structure factors were not accessible from the Protein Data Bank.

Growth signalling and binding studies were made with various receptor mutants and the SP2 isoform which lacks the N-terminal domain. The structures of the IL-3: IL-3 receptor alpha complexes were used to gain insights into the function of the IL-3 receptor. This approach has a major limitation as the most relevant structure for reference is the ternary complex which includes the beta common receptor. This receptor is present in the high affinity binding and growth assays used. As discussed in the text, previous binding studies have shown that a number of mutations which abolish direct binding with the alpha receptor show normal high affinity binding in the presence of the beta receptor indicating significant differences in the cytokine: alpha receptor interactions between the binary and ternary complexes. This limitation needs to be discussed.

Molecular dynamic simulations were used to predict mobility of the N-terminal domain of the receptor in the binary complex with IL-3. The hypothesis advanced in the paper is that binding of IL-3 to site 1 of the alpha receptor is the first and major step in the initiation of signalling, and that this limits the mobility of the N-terminal domain. This domain is suggested to prevent spontaneous alpha:beta heterodimerisation protecting cells from unregulated signalling.

There is quite a lot of data in the paper not consistent with this proposal. Firstly IL-3 alpha mutants with no direct binding of IL-3 gave high affinity binding and signalling so initial binding to site 1 is not obligatory for presentation to beta. In addition, although some spontaneous heterodimerisation was demonstrated for mutant alpha receptors with mobile or absent N-terminal domains, in no case was receptor phosphorylation observed in the absence of IL-3 so receptor activation was still fully regulated. A more balanced discussion needs to be given.

Throughout the paper and in the title the IL-3K116W mutant is referred to as a superkine. This seems inappropriate as the data in the paper indicates that this mutant has the same high affinity binding and growth signalling capability as wild type IL-3. It is only with the cell line TF1 where it is somewhat more active than wild type IL-3 in binding and signalling. The explanation given in the paper is low numbers of IL-3 receptors in TF1 compared with the cell systems used in the standard assays. However, at least as far as binding is concerned, the K_d is independent of receptor number and the binding programs used in the present work depend on this. It seems more likely that the IL-3 receptor in TF1 is different (in sequence or post-translational modifications) to the standard receptor used in this work. This should be discussed.

Response to Reviewer #1

(1) *"The manuscript would be strengthened by discussion of a related phenomenon seen in the IFNAR1 D1 domain that appears to be 'locked down' upon IFN binding, see works from Schreiber and Garcia on this."*

Thank you for the suggestion. We now include commentary on these studies in the **Discussion** on page 18.

Additional references now included in the **Reference** section:

Structural linkage between ligand discrimination and receptor activation by type I interferons.
Thomas C, Moraga I, Levin D, Krutzik PO, Podoplelova Y, Trejo A, Lee C, Yarden G, Vleck SE, Glenn JS, Nolan GP, Piehler J, Schreiber G, Garcia KC. Cell. 2011 Aug 19;146(4):621-32

Dynamic Modulation of Binding Affinity as a Mechanism for Regulating Interferon Signaling.
Li H, Sharma N, General IJ, Schreiber G, Bahar I. J Mol Biol. 2017 Aug 4;429(16):2571-2589.

(2) *"it is surprising that the authors use the term "superkine" without referencing the original use of the term from Levin et al., Nature 2012."*

This was an oversight and we have now included the citation in the **Introduction** on page 4.

Response to Reviewer #2

(1) *"In the main text, the results are simply referred to "MD simulations". Since molecular dynamics is a method, not a model, the term "MD simulations" is quite vague. For example, one could perform MD simulations using a coarse-grained model, rather than an all-atom model. At first usage, the authors should indicate that these are MD simulations with an all-atom explicit-solvent model."*

We agree with the reviewer and have now clarified that we use an all-atom explicit-solvent model for the MD simulations on page 5 of the manuscript.

(2) *“The manuscript misuses the term “stability”. That is, the section repeatedly describes the differences in the simulations as indicators of stability, though free-energy differences are not being evaluated. The authors should use more descriptive terms that reflect the quantities that are being calculated. It appears that the only analysis provided is time traces of interatomic distances. The claims of the main text (as written) are very broad (e.g. changes in stability) and are completely unsupported by the accompanying figures. It would be helpful if the claims are more precisely stated.”*

We agree with the reviewer and throughout the manuscript we have now replaced the term “stability” with terms that more precisely describe the calculated quantities and which support the accompanying figures. For example on pages 12 & 13 the sentence “Interestingly, MD simulations demonstrate that the IL-3 K116W binary complex has an increased stability compared to the wild-type IL-3 binary complex (**Fig. 5a-d, Supplementary Fig. 7h,i**),” now reads “Interestingly, the smaller root-mean-square fluctuation (RMSF) of the cytokine NTD C α atoms and the reduced interatomic distance between IL-3 E43 – IL3R α K54 and IL-3 D44 – IL3R α Y58 demonstrates that the IL-3 K116W binary complex has a reduced mobility compared to the wild-type IL-3 binary complex (**Fig. 5, Supplementary Fig. 7h,i**).”

Given the restriction in the number of allowed figures we felt that, along with the visual presentation of data in **Figure 6a-c** and **Supplementary Figure 4b**, the time traces of interatomic distances were the best way to illustrate the conformational differences between the (a) wild-type IL-3 and K116W mutant cytokine, (b) wild-type and mutant apo-IL3R α and (c) wild-type IL-3:IL3R α and IL-3 K116W:IL3R α binary complexes.

(3) *“While the methods state that simulations were performed for 50-200ns, figure S4 shows a 10ns trajectory and the main text interprets the time trace as indicating changes in confinement. Accordingly, it is unclear whether the simulations were actually 50-200ns, or not. A more general issue with this description is that the analysis does not appear to be systematic. For example, figure S4 is supposed to show that one system is more conformationally restricted, but the simulation appears to be only a few nanoseconds. Inspection of the traces in S4 show that both exhibit large fluctuations. Thus, it is not clear that the differences are beyond the uncertainty due to sampling limitations.”*

The production simulations were performed for 75 – 200 ns and not 50 - 200 ns as original stated in the **Online Methods** section; we have now amended this accordingly. In addition, we now clearly state the simulation timeframe for each system in either the manuscript text or the figure legend. When preparing the original figure panels, there was an error converting the number of frames to a ns time scale and this resulted in the X-axis time scale for **Figure 6d,e, Supplementary Figure 4c** and **Supplementary Figure 7a-i** being incorrect. The conversion error has now been rectified and the correct time scale is shown in ns. In addition, an early draft of **Supplementary Figure 4c** showing only 100 ns of simulation data was inserted into the manuscript by mistake – the data for 200 ns has now been included. When preparing **Supplementary Figure 7**, there was a cut and

paste error in panel **f** and the **g** panel image was inserted by mistake; the correct panel image has now been inserted into **Supplementary Figure 7f**.

Response to Reviewer #3

(1) *“For crystallization, an N212Q mutant of the receptor was used. It is important to state if this mutation affects binding or signalling. Fig1b (suppl) shows significant levels of impurities in the K116W complex. It is essential to verify these are absent from the crystals.”*

As suggested by the reviewer we have now made and tested the N212Q mutation and confirmed that it did not affect IL-3 binding and function. These new data are now included in the **Online Methods** section (page 23), **Supplementary Analysis** section (page 3) and **Supplementary Figure 3c,d**. Inclusion of the N212Q data has also required additional data to be included in Tables 2 and 3. The location of N212 in IL3R α is now shown in **Fig. 1a**.

We now include an additional image in **Supplementary Figure 1b**, that shows the protein samples used for crystallography. After fractions were pooled from the size exclusion column, they were buffer exchanged, concentrated and filtered through a 0.2 μ M filter. The new panel image shows the resultant protein from two different purifications and the absence of significant levels of impurity. An additional sentence clarifying this has also been included in the **Supplementary Figure 1b** legend.

(2) *“The sequence of the alpha receptor is given. What sequence was used for the SP2 isoform?”*

We now provide detailed information about the sequence of the IL3R α SP2 isoform within **Supplementary Figure 1c**.

(3) *“Data availability: The atomic coordinates and structure factors were not accessible from the Protein Data Bank.”*

The atomic coordinates and structure factors have been deposited in the Protein Data Bank and will be released upon publication of the manuscript. The PDB validation reports for both structures accompanied our submission.

(4) *“This approach has a major limitation as the most relevant structure for reference is the ternary complex which includes the beta common receptor. This receptor is present in the high affinity binding and growth assays used. As discussed in the text, previous binding studies have shown that a number of mutations which abolish direct binding with the alpha receptor show normal high affinity binding in the presence of the beta receptor indicating significant differences in the cytokine: alpha receptor interactions between the binary and ternary complexes. This limitation needs to be discussed.”*

We agree that these data have to be interpreted with caution in the absence of a structure for the IL-3 ternary receptor complex (IL-3:IL3R α : β c). Indeed, IL3R α mutants that do not measurably bind IL-3 in the absence of β c (direct binding), are able to bind IL-3 strongly when co-expressed with β c. Considering that β c does not measurably bind IL-3 on its own, we interpret this to mean that the IL3R α mutants had diminished binding, below the level of detection, but not ablated IL-3 recognition. We have now clarified and expanded the text on page 8 where we describe the restoration of IL-3 binding to IL3R α mutants in the presence of β c noting that "**The IL3R α NTD provides important interactions for optimal IL-3 binding and signalling**". Furthermore, we cannot completely exclude the possibility that the binding of IL-3 to a mutant IL3R α gives rise to a variant ternary complex that binds IL-3 with high-affinity but has slightly altered signalling properties. We have also included this possibility in the modified text (page 8).

(5) *"Molecular dynamic simulations were used to predict mobility of the N-terminal domain of the receptor in the binary complex with IL-3. The hypothesis advanced in the paper is that binding of IL-3 to site 1 of the alpha receptor is the first and major step in the initiation of signalling, and that this limits the mobility of the N-terminal domain. This domain is suggested to prevent spontaneous alpha:betac heterodimerisation protecting cells from unregulated signalling. There is quite a lot of data in the paper not consistent with this proposal. Q1. Firstly IL-3 alpha mutants with no direct binding of IL-3 gave high affinity binding and signalling so initial binding to site 1 is not obligatory for presentation to betac. Q2. In addition, although some spontaneous heterodimerisation was demonstrated for mutant alpha receptors with mobile or absent N-terminal domains, in no case was receptor phosphorylation observed in the absence of IL-3 so receptor activation was still fully regulated. A more balanced discussion needs to be given."*

Q1. As mentioned above, IL3R α mutants that do not measurably bind IL-3 in the absence of β c are able to bind IL-3 strongly when co-expressed with β c. Considering that β c does not measurably bind IL-3 on its own, we interpret this to mean that the IL3R α mutants had diminished but not ablated IL-3 recognition. Thus our interpretation of this data is that the Site 1 interaction is essential as the first and major step in the initiation of signalling even when the affinity of the Site 1 interaction is too low to be measured directly. We have now clarified the text (page 8) where we describe the restoration of IL-3 binding to IL3R α mutants in the presence of β c "**The IL3R α NTD provides important interactions for optimal IL-3 binding and signalling**".

Q2. We agree with the reviewer and we have now balanced the interpretation of this result by quoting and commenting on 2 new references (see below) of preformed receptor dimers that do not signal in the absence of ligand and on other ligand-induced conformational/translational change models related to receptor activation in this setting (**Discussion** section page 19).

Additional references now included in the **Reference** section.

Brown, R.J. et al. Model for growth hormone receptor activation based on subunit rotation within a receptor dimer. NSMB 12, 814-821 (2005).

Seubert, N. et al. Active and inactive orientations of the transmembrane and cytosolic domains of the erythropoietin receptor dimer. Mol. Cell 12, 1239-1250 (2003).

(6) “ Q1. Throughout the paper and in the title the IL-3K116W mutant is referred to as a superkine. This seems inappropriate as the data in the paper indicates that this mutant has the same high affinity binding and growth signalling capability as wild type IL-3. It is only with the cell line TF1 where it is somewhat more active than wild type IL-3 in binding and signalling. The explanation given in the paper is low numbers of IL-3 receptors in TF1 compared with the cell systems used in the standard assays. However, at least as far as binding is concerned, the Kd is independent of receptor number and the binding programs used in the present work depend on this. Q2. It seems more likely that the IL-3 receptor in TF1 is different (in sequence or post-translational modifications) to the standard receptor used in this work. This should be discussed.”

Q1. We agree with the reviewer and have now modified the text throughout the manuscript to refer to this IL-3 mutant by its specific mutation, i.e. IL-3 K116W. We also agree with the reviewer that the most likely explanation is that the endogenous IL-3 receptor in these TF1 cells may be somewhat different. As requested this is now discussed in page 11.

Additional reference now included in the **Reference** section.

Oon, S et al. Cytotoxic anti-IL-3R α antibody targets key cells and cytokines implicated in systemic lupus erythematosus. JCI Insight. 2016;1(6):e86131.

Q2. We agree that this is possible and it is now discussed in the penultimate paragraph of the **Discussion** (page 20), also noting that TF-1 cells are known to express both the wild-type full length SP1 and the truncated SP2 isoforms of IL3R α , at least at the level of mRNA (Ref 24 i.e. Chen, J. et al. A new isoform of interleukin-3 receptor {alpha} with novel differentiation activity and high affinity binding mode. J. Biol. Chem., **284**, 5763-73 (2009)).

REVIEWERS' COMMENTS:

Reviewer #1 (Remarks to the Author):

The revised MS has addressed my concerns.

Reviewer #2 (Remarks to the Author):

The authors have adequately addressed the technical points raised in the previous review. As a result, the description of the simulation details are now clear.

Reviewer #3 (Remarks to the Author):

The revised manuscript satisfactorily addresses the queries raised in the original review.